# Cost-effectiveness of psychological treatments for post-traumatic stress disorder in adults

**Ifigeneia Mavranezouli**[1,2]*, **Odette Megnin-Viggars**[1,2], **Nick Grey**[3,4], **Gita Bhutani**[5,6], **Jonathan Leach**[7], **Caitlin Daly**[8], **Sofia Dias**[8¤], **Nicky J. Welton**[8], **Cornelius Katona**[9,10], **Sharif El-Leithy**[11], **Neil Greenberg**[12], **Sarah Stockton**[2], **Stephen Pilling**[1,2,13]

**1** Centre for Outcomes Research and Effectiveness, Research Department of Clinical, Educational & Health Psychology, University College London, London, United Kingdom, **2** National Guideline Alliance, Royal College of Obstetricians and Gynaecologists, London, United Kingdom, **3** Sussex Partnership NHS Foundation Trust, Hove, United Kingdom, **4** School of Psychology, University of Sussex, Brighton, United Kingdom, **5** Lancaster & South Cumbria NHS Foundation Trust, Preston, United Kingdom, **6** University of Liverpool, Liverpool, United Kingdom, **7** Davenal House Surgery, Bromsgrove, United Kingdom, **8** Population Health Sciences, Bristol Medical School, University of Bristol, Bristol, United Kingdom, **9** Helen Bamber Foundation, London, United Kingdom, **10** Division of Psychiatry, University College London, London, United Kingdom, **11** Traumatic Stress Service, Springfield Hospital, London, United Kingdom, **12** King's Centre for Military Health Research, King's College London, London, United Kingdom, **13** Camden and Islington NHS Foundation Trust, St Pancras Hospital, London, United Kingdom

¤ Current address: Centre for Reviews and Dissemination, University of York, York, United Kingdom
* i.mavranezouli@ucl.ac.uk

## Abstract

### Background

Post-traumatic stress disorder (PTSD) is a severe and disabling condition that may lead to functional impairment and reduced productivity. Psychological interventions have been shown to be effective in its management. The objective of this study was to assess the cost-effectiveness of a range of interventions for adults with PTSD.

### Methods

A decision-analytic model was constructed to compare costs and quality-adjusted life-years (QALYs) of 10 interventions and no treatment for adults with PTSD, from the perspective of the National Health Service and personal social services in England. Effectiveness data were derived from a systematic review and network meta-analysis. Other model input parameters were based on published sources, supplemented by expert opinion.

### Results

Eye movement desensitisation and reprocessing (EMDR) appeared to be the most cost-effective intervention for adults with PTSD (with a probability of 0.34 amongst the 11 evaluated options at a cost-effectiveness threshold of £20,000/QALY), followed by combined somatic/cognitive therapies, self-help with support, psychoeducation, selective serotonin reuptake inhibitors (SSRIs), trauma-focused cognitive behavioural therapy (TF-CBT), self-

**Data Availability Statement:** Full details on the methods and the clinical studies included in the network meta-analysis that informed the economic analysis are provided in Mavranezouli et al.,

Psychol Med. 2020 Mar;50(4):542-555. doi: 10. 1017/S0033291720000070. All other relevant data are within the paper and its Supporting Information files.

**Funding:** The economic analysis presented in this paper was initiated by the National Collaborating Centre for Mental Health (NCCMH) and continued by the National Guideline Alliance (NGA) at the Royal College of Obstetricians and Gynaecologists (RCOG) from 1 April 2016, with support from the National Institute of Health and Care Excellence (NICE) Guidelines Technical Support Unit (TSU), University of Bristol, which is funded by the Centre for Guidelines (NICE). NCCMH and NGA received funding from NICE to develop clinical and social care guidelines. IM, OMG, SS and SP received support from the National Collaborating Centre for Mental Health and the National Guideline Alliance, which were in receipt of funding from the National Institute for Health and Care Excellence (NICE), for the submitted work. CD, SD and NJW received support from the NICE Guidelines Technical Support Unit, University of Bristol, with funding from the Centre for Clinical Practice (NICE). The views expressed in this publication are those of the authors and not necessarily those of the RCOG, NGA, NCCMH or NICE. The funder had no role in study design, data collection and analysis, decision to publish, or preparation of the manuscript. All authors had full access to all the data in the study and had final responsibility for the decision to submit for publication. National Institute for Health and Care Excellence (2018) Post-traumatic stress disorder. Available from: https://www.nice.org.uk/guidance/ng116

**Competing interests:** IM, OMV, SS and SP received support from the National Collaborating Centre for Mental Health and the National Guideline Alliance, which were in receipt of funding from the National Institute for Health and Care Excellence (NICE), for the submitted work. CD, SD and NJW received support from the NICE Guidelines Technical Support Unit, University of Bristol, with funding from the Centre for Clinical Practice (NICE). SD and NJW were co-applicants on a grant (unrelated to this work) from the MRC Methodology Research Programme which included an MRC Industry Collaboration Agreement with Pfizer Ltd, who part-funded a researcher to work on statistical methodology in a project underlated to this work. This does not alter our adherence to PLOS ONE policies on sharing data and materials. GB is a coinvestigator on a NIHR RfPB grant, Eye Movement Desensitization and Reprocessing Therapy in Early Psychosis (EYES): A feasibility randomised controlled trial.

help without support, non-TF-CBT and combined TF-CBT/SSRIs. Counselling appeared to be less cost-effective than no treatment. TF-CBT had the largest evidence base.

## Conclusions

A number of interventions appear to be cost-effective for the management of PTSD in adults. EMDR appears to be the most cost-effective amongst them. TF-CBT has the largest evidence base. There remains a need for well-conducted studies that examine the long-term clinical and cost-effectiveness of a range of treatments for adults with PTSD.

## Introduction

A considerable proportion of people exposed to trauma, around 5.6%, will develop post-traumatic stress disorder (PTSD) [1]. PTSD is a severe and disabling condition that may lead to functional impairment and reduced productivity [2]. A number of psychological interventions have been shown to be effective in the treatment of PTSD in adults, predominantly eye movement desensitisation and reprocessing (EMDR) and trauma-focused cognitive behavioural therapy (TF-CBT) [3]. However, many people with PTSD delay seeking help or are not identified by health services [4]. Given the variety of available interventions and the need for efficient use of healthcare resources, the objective of this study was to examine the cost-effectiveness of a range of psychological interventions for the treatment of PTSD in adults from the perspective of the National Health Service (NHS) and Personal Social Services (PSS) in England, using decision-analytic economic modelling.

The analysis presented here is part of the work that informed the updating of national guidance for the management of PTSD in England, published by the National Institute for Health and Care Excellence (NICE) [5]. The guideline was developed by a guideline committee, an independent multi-disciplinary group of clinical academics, health professionals and service user and carer representatives with expertise and experience in the field of PTSD. The committee contributed to the development of the economic model by providing advice on issues relating to the natural history and treatment patterns of PTSD in the UK, and on model inputs in areas where evidence was lacking.

## Methods

### Population

The study population comprised adults presenting in primary care with clinically important post-traumatic stress symptoms, defined by a diagnosis of PTSD according to the Diagnostic and Statistical Manual of Mental Disorders (DSM), the World Health Organization (WHO) International Classification of Diseases (ICD) or similar criteria, or by clinically significant PTSD symptoms, indicated by a PTSD symptom score above threshold on a validated scale, that are present for more than 3 months after a traumatic event.

The starting age of the cohorts in the economic model was 39 years, to reflect the mean age of adults with PTSD presenting to healthcare services in the UK [6]. The percentage of women in each cohort at the start of the model was 51.6%, calculated using national statistics for the general population [7], and data on the percentage of people screened positive for PTSD by age and sex in England [4]. The starting age and gender mix of the cohorts was used to estimate mortality risks and gender-specific quality-adjusted life-years (QALYs).

NGreenberg is the Royal College of Psychiatrists Lead for Military and Veterans' Health and is a trustee of two military charities. He is also a senior researcher with King's College London working on a number of military mental health r studies. NGrey is a member of the Wellcome Trust Anxiety Disorders Group developing, testing and disseminating Cognitive Therapy for PTSD (CT-PTSD), a trauma-focused cognitive behavioural therapy (TF-CBT). He has published papers and book chapters on CT-PTSD, and facilitates teaching workshops for which payment is received. As editor, he receives royalties from sales of a trauma book, A Casebook of Cognitive Therapy for Traumatic Stress Reactions. CK is Medical Director of the Helen Bamber Foundation (a human rights charity) and refugee and asylum mental health lead for the Royal College of Psychiatrists. He writes expert psychiatric reports in the context of asylum mental health. JL is NHS England Medical Director for Military and Veterans Health. SP receives funding from NICE for the development of clinical guidelines and is also supported by the NIHR UCLH Biomedical Research Centre. The authors report no other relationships or activities that could appear to have influenced the submitted work.

## Interventions

The interventions considered in the economic analysis were selected from those considered in a network meta-analysis (NMA) of randomised controlled trials (RCTs) of psychological treatments for adults with PTSD ([3]; see S2 Appendix for inclusion criteria for the NMA). We included only interventions that had been tested on at least 100 individuals in the NMA of changes in PTSD symptoms at treatment endpoint, as this was deemed the minimum adequate evidence base that would enable robust conclusions to be drawn on clinical and cost-effectiveness. Moreover, we included only interventions that had shown a higher mean effect in comparison with waitlist. Selective serotonin reuptake inhibitors (SSRIs) were included in the analysis as they were relevant comparators to psychological interventions.

The economic analysis evaluated the following interventions:

- EMDR

- TF-CBT

- Non-TF-CBT

- Combined somatic/cognitive therapies

- SSRIs

- Combined TF-CBT/SSRIs

- Self-help with support

- Self-help without support

- Counselling

- Psychoeducation

- No treatment, reflected in waitlist RCT arms.

TF-CBT is a broad class of psychological interventions that predominantly use trauma-focused cognitive, behavioural or cognitive-behavioural techniques and exposure approaches to treatment. Although some interventions place their main emphasis on exposure and others on cognitive techniques, most use a combination of both. TF-CBT includes therapies such as cognitive therapy, cognitive processing therapy, exposure therapy/prolonged exposure, virtual reality exposure therapy, mindfulness-based cognitive therapy and narrative exposure therapy. Although the specific interventions that make up a class do not include exactly the same content or follow the same manual, they use the same broad approach and there is considerable overlap in the proposed mechanisms. In the economic analysis that informed the NICE guideline [5] we divided the TF-CBT class by the number of sessions and format of delivery and created separate categories of TF-CBT treatment according to their intensity, as these differences in resource use comprised practical considerations that informed the guideline recommendations; in addition to different intervention costs, each TF-CBT category had its own clinical effectiveness, estimated in the guideline NMAs. However, in the analyses we present here, we considered TF-CBT as an umbrella term of interventions that share a similar approach to treatment, in order to investigate the overall performance of the TF-CBT class relative to other treatments, regardless of its mode of delivery. Resource use for TF-CBT in the economic analysis we present here was determined by the average resource use reported in the TF-CBT trials informing the analysis, considering also that their vast majority assessed individual forms of TF-CBT.

Non-TF-CBT is a class of interventions that focus on current symptoms of PTSD without re-visiting the trauma experience. Combined somatic/cognitive therapies, such as emotional freedom techniques and thought field therapy, are exposure-based therapies with both cognitive and somatic components that utilise the tapping of points on the body [8, 9]. EMDR is based on a theoretical model which posits that the dysfunctional intrusions, emotions and physical sensations experienced by trauma victims are due to the improper storage of the traumatic event in implicit memory. The EMDR procedures are based on stimulating the patient's own information processing in order to help integrate the targeted event as an adaptive contextualised memory [10]. Counselling is a type of psychological treatment that builds on the concept of client-centred therapy by Rogers [11]. It has been described in the relevant literature as non-directive counselling, supportive counselling, supportive psychotherapy, or person-centred counselling. Individuals are helped to focus on their thoughts, feelings and behaviour; to reach clearer self-understanding; and to find and use their strengths so that they cope more effectively with their lives by making appropriate decisions, or by taking relevant action. Counselling is primarily non-directive and non-advisory, but recognises that some situations require positive guidance by means of information and advice. Psychoeducation involves the provision of information about the nature and causes of PTSD, and strategies and treatments that can help address PTSD symptoms. It can be delivered individually, but is commonly delivered to groups. Psychoeducation is usually non-directive and takes an educational didactic format. Finally, self-help therapies include interventions such as internet-based TF-CBT or other computerised psychological therapies, expressive writing and cognitive bibliotherapy. Self-help with support includes interventions in which therapist's input is an integral part of the intervention; in self-help without support the therapist's input is minimal or absent.

The guideline analysis included interventions tested on at least 50 people in the NMA of changes in PTSD symptoms at treatment endpoint, whereas here we used a threshold of at least 100 people to improve the robustness of the results. The impact of increasing this threshold was the omission of interpersonal psychotherapy and present-centered therapy from the analysis presented here; both interventions were shown to occupy middle-to-lower cost-effectiveness rankings in the guideline analysis, and therefore their omission had no impact on the overall conclusions of our analysis.

## Economic model structure

A hybrid decision-analytic model consisting of a decision-tree followed by a three-state Markov model was constructed using Microsoft Office Excel 2013 to estimate total costs and QALYs associated with each treatment. The model structure was determined by the natural history of PTSD, its treatment patterns in the UK, and the availability of relevant clinical and epidemiological data (Fig 1).

The model followed hypothetical cohorts of adults with PTSD, initiated on each of the treatments assessed. The treatment duration for each of the assessed options equalled 3 months (12 weeks), according to the average duration of interventions in trials and routine clinical practice (range 4–20 weeks). Following a course of treatment, people in each cohort either remitted (entering a state of 'no-PTSD') or failed to remit, remaining in a 'PTSD' state. Those initiated on SSRIs alone or in combination were given 3 months of maintenance pharmacological therapy if they had remitted. Death was not considered during provision of interventions, as no relevant differential mortality data are available. In the next 3 months of follow-up, those who had remitted could remain in remission, relapse to PTSD or die. Those who had not remitted could remain in the 'PTSD' state, remit (and move to 'no-PTSD') or die. The length of the

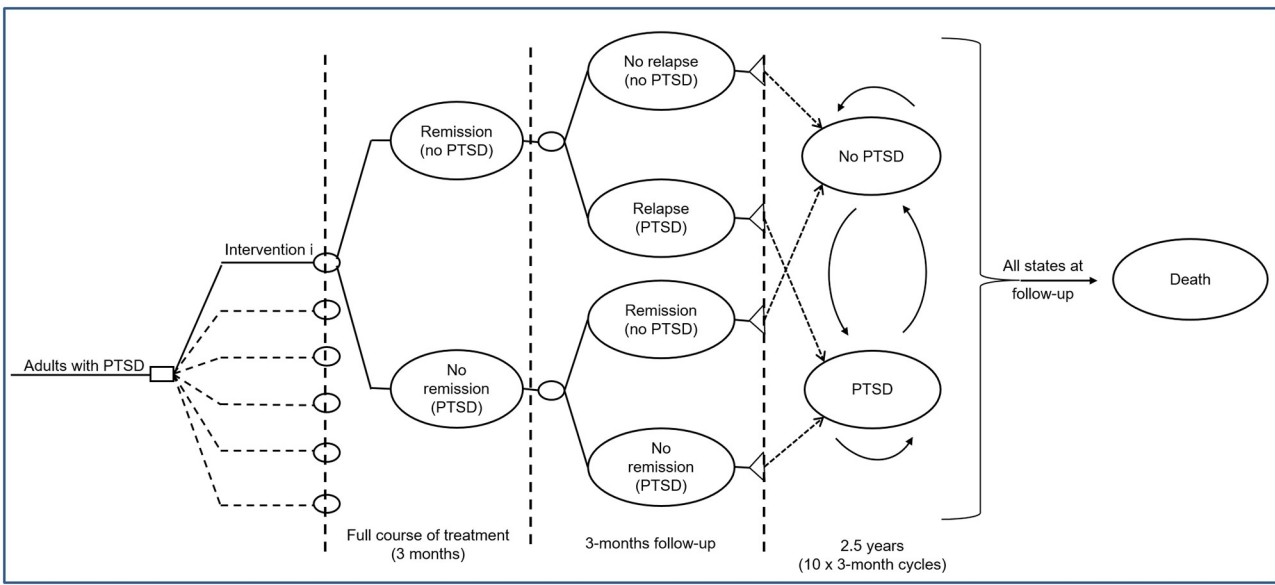

**Fig 1. Schematic diagram of the economic model.**

follow-up period immediately post-treatment was set at 3 months as this is the period for which most follow-up data are reported in RCTs of psychological interventions for PTSD.

After that point, people in each cohort entered the Markov model, run in 3-month cycles, for consistency with the duration of the two periods of the decision-tree. In each cycle, they remained in the same health state or moved between the states of 'PTSD' and 'no-PTSD' or moved to death (absorbing state). People in both the 'PTSD' and the 'no-PTSD' states received primary, community and secondary healthcare and personal social services, as relevant to their health state. A half-cycle correction was applied.

The time horizon of the analysis was 3 years (36 months), comprising 6 months in the decision tree and 2.5 years (10 x 3-month cycles) in the Markov component of the model. This time frame was deemed adequate to capture longer-term costs and effects of treatment, without making significant extrapolations and assumptions over the course of PTSD.

## Effectiveness data

We obtained effectiveness data from a systematic review and NMA of psychological treatments for adults with PTSD [3]. We utilised the results of 2 NMAs of changes in PTSD symptoms: between baseline and treatment endpoint; and between baseline and 1-4-month follow-up. Details on the NMAs, including the studies and data that informed them, the selection of the effectiveness data and the transformations required for use in the economic model are provided in S1 File.

The outputs of the NMA of changes in PTSD symptoms between baseline and treatment endpoint informed the intervention effects in the model period of 0–3 months. For the 3–6 month follow-up period, the base-case economic analysis conservatively assumed that the active intervention effects were not retained and equalled the effect of no treatment; this was decided because the results of the NMA of changes in PTSD symptoms between baseline and 1-4-month follow-up were based on limited evidence and showed considerable uncertainty. Nevertheless, data from this NMA were used in a secondary analysis, to inform effects for each active intervention during 3–6 months after treatment initiation.

### Baseline probability of remission

The probability of remission for no treatment (baseline) and for all model arms beyond treatment endpoint (i.e. for all treatment options during 3–6 months after treatment initiation in the base-case analysis and for all treatment options during 6–36 months after treatment initiation in both the base-case and secondary analysis) was estimated using data from an Australian national mental health survey [12]. We considered the survey participants to be representative of our study population, which was adults presenting in primary care with symptoms of PTSD. Details on the methods used for the estimation of the baseline probability of remission are provided in S2 File.

### Risk of relapse

Due to lack of published evidence, an annual risk of relapse of 0.10 was assumed across all treatment arms, based on the committee's expert opinion; this was translated into a 3-month probability of relapse of 0.026 assuming an exponential function, which was applied in the 3-month follow-up period of the decision-tree and over the whole duration of the Markov model. This assumption was tested in a sensitivity analysis.

### Risk of development of side effects from SSRI treatment

The probability of developing common side effects from SSRIs (headaches, nausea or vomiting, agitation, sedation and sexual dysfunction) was estimated from a retrospective analysis of a large US managed care claims database on 40,017 people with depression who were initiated on antidepressant monotherapy, including SSRIs [13]. Serious side effects from SSRIs (such as death, attempted suicide or self-harm, falls, fractures, stroke, epilepsy/seizures) were not considered; however, their omission is not expected to have had a significant impact on the economic results, due to their low incidence in the study population [14, 15].

### Mortality

PTSD is associated with an increased risk of mortality. The hazard ratio of death associated with PTSD, adjusted for confounders such as age, gender, diabetes mellitus and hypertension was obtained from a study on 637 veterans in the US [16]. This ratio was applied onto general mortality statistics for England [17] to estimate the annual mortality risk in people with PTSD over the time period they experienced PTSD symptoms. People without PTSD symptoms had the mortality risk of the general population.

### Utility data

Utility scores express preferences for the health-related quality of life (HRQoL) in distinct health states; they are necessary for the estimation of QALYs. Following a systematic literature search of utility data for PTSD, the base-case economic analysis used utility scores generated from HRQoL ratings of Australian adult participants in a national mental health survey, some of whom had a diagnosis of PTSD according to DSM-4 criteria [18]. HRQoL was assessed with the Assessment of Quality of Life measure (http://www.aqol.com.au). The study provided gender-specific data for people with PTSD and people who were PTSD-free following evidence-based treatment, which corresponded directly to the model health states of interest.

In a sensitivity analysis, we used utility data derived from a sample of 808 US veterans attending primary care clinics, 97 of whom had developed PTSD, adjusted for confounders such as gender, employment status, presence of disability, and mental and physical health comorbidities [19]. HRQoL was assessed using SF-36, which was converted to utility scores

using the UK algorithm [20]. Data from this study indicated a narrower utility benefit following remission from PTSD compared with the utility data used in the base-case analysis.

The mean utility decrement in people experiencing common side effects from SSRIs was estimated using EQ-5D scores of participants in a US medical expenditure survey, some of whom had depression and experienced side effects from antidepressant treatment [21]. This utility decrement was applied only over the period that people received SSRI treatment.

## Resource use and cost data

The analysis included intervention costs (healthcare professional time, drug acquisition and equipment/infrastructure required for self-help interventions), costs of managing side effects from medication, and costs relating to the 'PTSD' and 'no-PTSD' health states including costs of primary, community and secondary healthcare and PSS costs.

Intervention costs (Table 1) were calculated by combining resource use reported in RCTs included in the NMA that informed the economic analysis (i.e. number and duration of therapeutic sessions and mean daily dosage of sertraline, which was the most commonly used SSRI in trials), modified to represent routine UK practice, with respective national unit costs. All psychological interventions, with the exception of self-help and psychoeducation, were assumed to be delivered in a primary care setting by Band 7 psychological therapists according to the NHS Agenda for Change (AfC) pay-scales for community-based scientific and professional staff, to reflect routine practice in the UK. Psychoeducation was assumed to be delivered by AfC band 5 Psychological Well-being Practitioners (PWPs) and self-help by AfC band 6 psychological therapists. People receiving SSRIs attended general practitioner (GP) monitoring visits and undertook routine laboratory testing. Those experiencing side effects had one extra GP contact every 3 months and received medication for their management. For self-help therapies we included the cost of the provider of digital mental health programmes, and costs of hardware and capital overheads.

Unit costs were estimated using a combination of data derived from national sources and other published evidence [22–26]. Health professional unit costs included wages/salary, salary on-costs, capital and other overheads, qualification costs where available, and supervision costs. The ratio of direct (face-to-face) to indirect (preparation and administrative tasks) health professionals' time was taken into account. Details on the methods and sources used to estimate therapists' unit costs are reported in S3 File.

Annual costs associated with the PTSD and no-PTSD health states were estimated using predominantly NHS and PSS usage data from a national psychiatric morbidity survey conducted in England in 2014 [4], supplemented with resource use data from other published sources [27–29] and expert opinion, which were subsequently combined with national unit costs [22, 23, 30]. Costs for each state included inpatient hospital stays and outpatient visits, contacts with GPs, psychiatrists, psychologists, social workers, community psychiatric and learning disability nurses, other nursing services, self-help and support groups, home help or home care, outreach or family support workers and community day-care centres. Details on the methods and data used to estimate annual costs associated with the PTSD and no-PTSD health states are provided in S4 File. These were then translated into 3-month cost figures that informed the economic model. Because the estimated health state-related costs were based to a large extent on expert opinion, a sensitivity analysis was conducted, in which PTSD costs were varied by ±50%.

Costs were expressed in 2017 prices, uplifted, where necessary, using the Hospital and Community Health Services Pay and Prices Index [22].

**Table 1. Intervention costs of treatments for adults with PTSD (2017 prices).**

| Intervention | Resource use details | Intervention cost |
|---|---|---|
| **EMDR** | 6 x 1.5 hr individual sessions (9 hours) delivered by a Band 7 psychological therapist | £912 |
| **TF-CBT** | 9 x 1.5 hr individual sessions (13.5 hours) delivered by a Band 7 psychological therapist | £1,368 |
| **non-TF-CBT** | 9 x 1 hr individual sessions (9 hours) delivered by a Band 7 psychological therapist | £912 |
| **Combined somatic/cognitive therapies** | 4 x 1 hr individual sessions (4 hours) delivered by a Band 7 psychological therapist | £405 |
| **SSRIs (sertraline)** | Mean daily dosage 150mg, 4 GP visits at 0–3 months + 1 visit at 3–6 months, monitoring lab testing | 0–3 months: £155 |
| | For people experiencing side effects: 1 extra GP visit over 3 months, medication for management. | 3–6 months: £39<br>If side effects: £37 |
| **Combined TF-CBT/SSRIs** | Sum of the individual treatment components | 0–3 months: £1,523 3–6 months: £39 |
| **Self-help with support** | Fixed cost of provider of digital programmes, hardware & capital overheads, 180 minutes of support by a band 6 psychological therapist | £266 |
| **Self-help without support** | Fixed cost of provider of digital programmes, hardware & capital overheads, 40 minutes of support by a band 6 psychological therapist | £98 |
| **Counselling** | 10 x 1 hr individual sessions (10 hours) delivered by a Band 7 psychological therapist | £1,014 |
| **Psychoeducation** | 3 x 1 hr individual sessions (3 hours) delivered by a band 5 PWP | £127 |
| **No treatment** | No resource use | £0 |

EMDR: eye movement desensitisation reprocessing; IPT: interpersonal psychotherapy; PWP: psychological well-being practitioner; TF-CBT: trauma-focused cognitive behavioural therapy

Unit cost of band 7 psychological therapists: £101 per hour of direct contact with the client–see S3 File for details on estimation of unit cost

Unit cost of band 5 PWPs: £42 per hour of direct contact with the client–see S3 File for details on estimation of unit cost

Unit cost of band 6 psychological therapist: £72 per hour of direct contact with the client (mean value of unit costs of band 7 psychological therapist and band 5 PWP)

Unit cost of GP: £37 per patient contact lasting 9.22 minutes [22]

Sertraline acquisition cost: 100mg, 28 tab, £0.99 [23]– 3-month cost £1.59

Cost of monitoring lab testing (SSRIs): £5 per person (expert advice)

Cost of medication for management of side effects (SSRIs): £3 per person over 3 months (expert advice)

Fixed cost of provider of digital programmes: £36.20 per person (expert advice)

Cost of hardware & capital overheads: £14 per person [24]

## Discounting

Costs and QALYs were discounted at 3.5% annually as recommended by NICE [31].

## Analysis

To account for the uncertainty around input parameter point estimates, a probabilistic analysis was undertaken, in which input parameters were assigned probabilistic distributions [32]. Subsequently, 10,000 iterations were performed, each drawing random values out of the distributions fitted onto the model input parameters. Mean costs and QALYs for each treatment were calculated by averaging across the 10,000 iterations. The Net Monetary Benefit (NMB) for each intervention was estimated for each iteration and averaged across the 10,000 iterations, determined by the formula

$$NMB = E \bullet \lambda - C$$

where E and C are the effects (QALYs) and costs of each intervention, respectively, and λ represents the willingness-to-pay per unit of effectiveness, set at the NICE lower cost-effectiveness threshold of £20,000/QALY [31]. The intervention with the highest NMB is the most cost-effective option [33].

The mean ranking by cost-effectiveness is reported for each intervention (out of 10,000 iterations), where a rank of 1 suggests that an intervention is the most cost-effective amongst all evaluated treatment options. The probability of the intervention with the highest NMB being the most cost-effective option is also provided, calculated as the proportion of iterations in which the intervention has had the highest NMB amongst all interventions considered in the analysis. The probability of cost-effectiveness has been estimated in a step-wise approach, according to which the most cost-effective intervention is omitted at each step, and the probability of cost-effectiveness of the next most cost-effective intervention amongst the remaining treatment options is re-calculated. The probabilities estimated following this approach reflect the uncertainty around the cost-effectiveness not only of the most cost-effective intervention, but also of the second, third, fourth, etc. most cost-effective intervention in ranking, after more cost-effective interventions have been omitted from analysis. Finally, the cost-effectiveness acceptability frontier has been plotted, showing the treatment with the highest mean NMB over different cost-effectiveness thresholds ($\lambda$), and the probability that this treatment is the most cost-effective among those assessed [33].

Table 2 reports the values of all model input parameters. Deterministic values were used in deterministic one-way sensitivity analyses. The probability distributions show the types and range of distributions assigned to each parameter; estimation of distribution ranges was based on data reported in the published sources of evidence.

Two probabilistic analyses were undertaken, each using different assumptions on the effectiveness of interventions at the 3-month follow-up:

- Base-case analysis: treatment effect between 3–6 months equalled that of no treatment

- Secondary analysis: treatment effect between 3–6 months equalled that estimated from the NMA of changes in PTSD symptoms between baseline and 1-4-month follow-up

One-way deterministic sensitivity analyses explored the following scenarios applied onto the base-case analysis:

- change in the annual risk of relapse between 0.05 and 0.20

- change in the PTSD health state cost by ± 50%

- use of alternative utility scores for the PTSD and no-PTSD states [19].

## Validation of the economic model

The economic model was developed in collaboration with members of the guideline committee. All inputs and model formulae were systematically checked. The model was tested for logical consistency by setting input parameters to null and extreme values and examining whether results changed in the expected direction. Results were discussed with the committee to confirm their plausibility.

## Results

Table 3 shows the results of the base-case economic analysis. Interventions have been ordered from the most to the least cost-effective. The table provides the mean number of QALYs, intervention costs and total costs per person, the mean NMB and ranking of each intervention, and its probability of being cost-effective in a step-wise approach at a threshold of £20,000/QALY.

EMDR was found to be the most cost-effective intervention for adults with PTSD, with the highest NMB at the cost-effectiveness threshold of £20,000/QALY. This was followed by

**Table 2. Economic model input parameters.**

| Input parameter | Deterministic value | Probability distribution (type, range) | Sources–comments |
|---|---|---|---|
| **Characteristics of study population** | | | |
| Starting age of cohort (years) | 39 | No distribution | [6]; mean age of adults referred for assessment for possible PTSD in a UK NHS outpatient clinic |
| Proportion of women | 0.52 | No distribution | Calculated using the proportion of women in the general population aged 39 years [7], and data on the percentage of people screened positive for PTSD by age and sex [4]. |
| **Odds ratios of remission versus no treatment at treatment endpoint** | | | |
| EMDR | 42.18 | 95% CrI: 13.59 to 132.42 | [3]; standardised mean differences converted to odds ratios according to [34]; distribution based on 300,000 samples from posterior distributions outputted from NMAs, thinned by 30 to obtain 10,000 values |
| TF-CBT | 14.06 | 95% CrI: 6.76 to 29.81 | |
| non-TF-CBT | 9.09 | 95% CrI: 2.50 to 33.62 | |
| Combined somatic/ cognitive therapies | 21.33 | 95% CrI: 3.84 to 121.63 | |
| SSRIs | 7.99 | 95% CrI: 1.50 to 44.61 | |
| Combined TF-CBT/SSRIs | 9.06 | 95% CrI: 1.15 to 69.34 | |
| Self-help with support | 13.98 | 95% CrI: 2.74 to 70.74 | |
| Self-help without support | 5.17 | 95% CrI: 1.29 to 20.27 | |
| Counselling | 3.70 | 95% CrI: 1.12 to 12.38 | |
| Psychoeducation | 8.99 | 95% CrI: 0.26 to 276.72 | |
| **Odds ratios of remission versus no treatment at 3-month follow-up (secondary analysis only)** | | | |
| EMDR | 7.53 | 95% CrI: 1.55 to 35.77 | [3]; standardised mean differences converted to odds ratios according to [34]; distribution based on 300,000 samples from posterior distributions outputted from NMAs, thinned by 30 to obtain 10,000 values |
| TF-CBT | 3.80 | 95% CrI: 1.49 to 9.72 | |
| non-TF-CBT | 2.18 | 95% CrI: 0.37 to 12.40 | |
| Combined somatic/ cognitive therapies | 8.08 | 95% CrI: 0.41 to 155.56 | 3-6-month probability of remission for SSRIs assumed to equal the probability of remission of SSRIs during initial treatment (0–3 months); 3-6-month probability of remission for combined TF-CBT/SSRIs borrowed from TF-CBT. |
| SSRIs | No data | No data | |
| Combined TF-CBT/SSRIs | No data | No data | |
| Self-help with support | 10.11 | 95% CrI: 2.03 to 48.96 | |
| Self-help without support | 8.85 | 95% CrI: 0.73 to 105.43 | |
| Counselling | 1.73 | 95% CrI: 0.37 to 8.15 | |
| Psychoeducation | 2.58 | 95% CrI: 0.42 to 15.43 | |
| **Probability of remission–no treatment (also applied to all interventions between 3–6 months in the base-case analysis & all interventions beyond 6 months in the base-case and secondary analyses)** | | | |
| 0–3 months from PTSD onset | 0.03 | Beta: $\alpha$ = 17.26; $\beta$ = 646.74 | [12]. See S2 File for details |
| 0–12 months from PTSD onset | 0.15 | Beta: $\alpha$ = 98.94; $\beta$ = 565.06 | |
| 0–24 months from PTSD onset | 0.27 | Beta: $\alpha$ = 176.62; $\beta$ = 487.38 | |
| 0–36 months from PTSD onset | 0.32 | Beta: $\alpha$ = 212.48; $\beta$ = 451.52 | |
| **Risk of relapse–all treatments** | | | |
| 3-month risk | 0.026 | Beta: $\alpha$ = 2.60; $\beta$ = 97.40 | Expert opinion |
| **Risk of developing common side effects from SSRIs** | | | |
| 3-month risk | 0.029 | Beta: $\alpha$ = 687; $\beta$ = 22,933 | [13] |
| **Mortality** | | | |
| Hazard ratio–PTSD vs no PTSD | 1.77 | Log-normal: 95% CI 1.02 to 3.14 | [16] |
| Baseline mortality– general population | Age/sex specific | No distribution | General mortality statistics for the UK population [17] |

(*Continued*)

**Table 2.** (*Continued*)

| Input parameter | Deterministic value | Probability distribution (type, range) | Sources–comments |
|---|---|---|---|
| **Utility values** | | | |
| Base-case analysis | | | |
| PTSD, men | 0.54 | Beta: α = 26.83; β = 22.86 | [18]; distribution estimated based on method of moments |
| PTSD, women | 0.57 | Beta: α = 86.75; β = 65.44 | |
| No PTSD, men | 0.63 | Beta: α = 5.11; β = 3.00 | |
| No PTSD, women | 0.64 | Beta: α = 14.11; β = 7.93 | |
| Sensitivity analysis | | | |
| PTSD, all | 0.61 | No distribution | [19] |
| No PTSD, all | 0.64 | | |
| Disutility due to side effects from SSRIs | | | |
| % reduction in health state utility | 10.3 | Beta: α = 89.64; β = 784.07 | [21]; disutility applied as a percentage onto the health state (PTSD or no PTSD) utility |
| **Intervention costs–resource use** | | | |
| Number of sessions | | | |
| EMDR | 6 | 0.70: 5–6, 0.16: 4, 0.14: 3 | Different probabilities assigned to different numbers of sessions for each therapy, based on resource use reported in the RCTs included in the NMAs that informed the economic analysis, supplemented by further assumptions. Combined TF-CBT/SSRIs: resource use was the sum of the resource use of the individual treatment components. Details on intervention costs are provided in Table 1. |
| TF-CBT | 9 | 0.70: 7–9, 0.16: 5–6, 0.14: 3–4 | |
| non-TF-CBT | 9 | 0.70: 7–9, 0.16: 5–6, 0.14: 3–4 | |
| Combined somatic/cognitive therapies | 4 | 0.70: 4, 0.30: 2–3 | |
| Counselling | 10 | 0.70: 8–10, 0.16: 5–7, 0.14: 3–4 | |
| Psychoeducation | 3 | 0.70: 3, 0.16: 2, 0.14: 1 | |
| Therapist time (minutes) | | | |
| Self-help with support | 180 | Normal: SD = 0.30*mean | SD based on assumption |
| Self-help without support | 40 | Normal: SD = 0.30*mean | |
| Number of GP contacts–SSRIs | | | |
| 0–3 months | 4 | 0.70: 4, 0.30: 2–3 | Different probabilities assigned to different numbers of sessions; number of visits based on expert opinion; distribution based on assumption. |
| 3–6 months | 1 | 0.70: 1, 0.30: 0 | |
| Treatment of side effects | 1 | 0.80: 1, 0.20: 2 | |
| **Intervention costs—unit costs** | | | |
| SSRI– 3-month drug acquisition | £1.59 | No distribution | [23] |
| Laboratory testing–SSRIs | £5 | No distribution | Assumption |
| Medication for side effects–SSRIs | £3 | No distribution | Assumption |
| Self-help infrastructure | £50 | No distribution | Fixed digital therapy provider cost based on expert advice; capital cost based on [24] |
| GP unit cost | £37 | Normal, SE = 0.05 of the mean | [22]; distribution based on assumption |
| Band 7 clinical psychologist unit cost | £101 | Normal: SE = 0.05 of the mean | See S3 File; distribution based on assumption |
| Band 5 PWP unit cost | £42 | Normal, SE = 0.05 of the mean | See S3 File; distribution based on assumption |
| Band 6 therapist unit cost | £72 | Determined by distribution of Band 7 and Band 5 therapist unit costs | Assumed to be the mean of Band 7 and Band 5 therapist unit costs |
| **3-month NHS/PSS health state cost** | | | |
| PTSD | £293 | Gamma: SE = 0.30 of the mean | Based on resource use data reported in national and other published sources [4, 27–29], supplemented with expert opinion and combined with national unit costs [22, 23, 30], expressed in 2017 prices; see S4 File for details. |
| No-PTSD | £27 | Gamma: SE = 0.30 of the mean | |

(*Continued*)

**Table 2.**  (Continued)

| Input parameter | Deterministic value | Probability distribution (type, range) | Sources–comments |
|---|---|---|---|
| Annual discount rate | 0.035 | No distribution | Applied to costs and QALYs [31] |

CI: confidence intervals; CrI: credible intervals; EMDR: eye movement desensitisation reprocessing; GP: general practitioner; IPT: interpersonal psychotherapy; NHS: National Health Service; PSS: personal social services; PTSD: post-traumatic stress disorder; PWP: psychological well-being practitioner; SD: standard deviation; SE: standard error; SSRI: selective serotonin reuptake inhibitor; TF-CBT: trauma-focused cognitive behavioural therapy

combined somatic/cognitive therapies, self-help with support, psychoeducation, SSRIs, TF-CBT, self-help without support, non-TF-CBT, combined TF-CBT/SSRIs, no treatment and counselling. The probability of EMDR being the most cost-effective treatment amongst the 11 options assessed was only 0.34. The probabilities of cost-effectiveness for next interventions in ranking up to (and including) self-help without support did not exceed 0.42, although increasingly fewer interventions were included in the analysis, indicating uncertainty in the results. Notably, counselling was less cost-effective than no treatment; this finding was attributed to the relatively low clinical effectiveness of counselling (the lowest amongst all active treatments assessed in the economic analysis), which did not offset its relatively high intervention cost. The cost-effectiveness plane (Fig 2) depicts the mean incremental costs and QALYs of all interventions versus no treatment (placed at the origin). According to the cost-effectiveness acceptability frontier (Fig 3), combined somatic/cognitive therapies appeared to be most cost-effective amongst the 11 treatment options assessed for thresholds up to £2,500/QALY, with a low probability that reached 0.25 at maximum, whereas EMDR became the most cost-effective option at higher thresholds, with a probability that ranged from 0.19 to 0.41 amongst the 11 options assessed.

Results of the secondary analysis, which utilised 3-month NMA follow-up data, were not very different. The top 3 interventions (EMDR, combined somatic/cognitive therapies, self-

**Table 3.  Base-case results of economic modelling.**

| Intervention | Mean per person | | | NMB (£/ person) | Mean rank | Prob* |
|---|---|---|---|---|---|---|
| | QALY | Intervention cost (£) | Total cost (£) | | (at a threshold of £20,000/QALY) | |
| EMDR | 1.80 | 746 | 2,047 | 33,928 | 2.31 | 0.34 |
| Combined somatic/cognitive therapies | 1.77 | 360 | 1,963 | 33,364 | 3.28 | 0.35 |
| Self-help with support | 1.75 | 266 | 2,047 | 32,880 | 4.01 | 0.32 |
| Psychoeducation | 1.74 | 108 | 1,982 | 32,754 | 4.90 | 0.42 |
| SSRIs | 1.72 | 146 | 2,143 | 32,316 | 5.15 | 0.37 |
| TF-CBT | 1.74 | 1,058 | 2,854 | 32,042 | 6.38 | 0.26 |
| Self-help without support | 1.71 | 98 | 2,253 | 31,865 | 6.19 | 0.41 |
| non-TF-CBT | 1.73 | 705 | 2,670 | 31,860 | 6.79 | 0.50 |
| Combined TF-CBT/SSRIs | 1.73 | 1,204 | 3,140 | 31,451 | 8.19 | 0.48 |
| No treatment | 1.67 | 0 | 2,488 | 30,915 | 9.14 | 0.64 |
| Counselling | 1.69 | 785 | 3,043 | 30,854 | 9.66 | 1.00 |

EMDR: eye movement desensitisation reprocessing; NMB: net monetary benefit; Prob: probability of cost-effectiveness; SSRIs: selective serotonin reuptake inhibitors; TF-CBT: trauma-focused cognitive behavioural therapy

*estimated in a step-wise approach, according to which the most cost-effective intervention is omitted at each step, and the probability of cost-effectiveness of the next most cost-effective intervention amongst the remaining treatment options is re-calculated

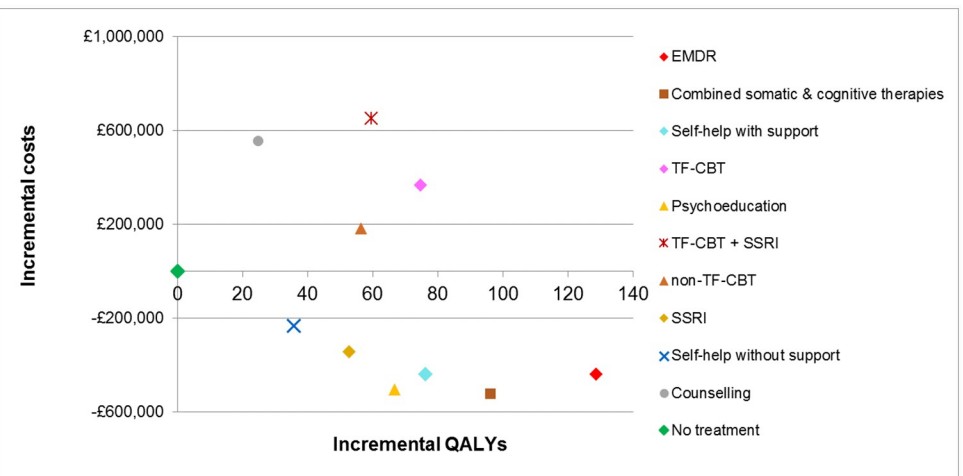

**Fig 2. Cost-effectiveness plane: Base-case analysis results for 1,000 adults with PTSD.**

help with support) remained the same. The ranking of self-help without support improved and counselling became better than no treatment. The probabilities of cost-effectiveness of the top 5 interventions were low, ranging between 0.14 and 0.47, indicating uncertainty around the results. Self-help without support appeared to be the most cost-effective option at a zero cost-effectiveness threshold and combined somatic/cognitive therapies were most cost-effective at higher thresholds up to £18,000/QALY; EMDR was the most cost-effective option at higher thresholds, with a 0.14 probability at the threshold of £20,000/QALY. Results of the secondary analysis are provided in S5 File.

In deterministic sensitivity analyses, results were, overall, robust to changes in the risk of relapse and in the PTSD health state cost and rankings were not affected. TF-CBT was the only option that dropped (by one place) in ranking when the baseline risk of relapse was increased by 50% or the PTSD health state cost was reduced by 50%. Use of alternative utility data that assumed narrower HRQoL benefits associated with remission had a small impact on the results, with the relative cost-effectiveness of TF-CBT alone or combined with SSRIs and non-

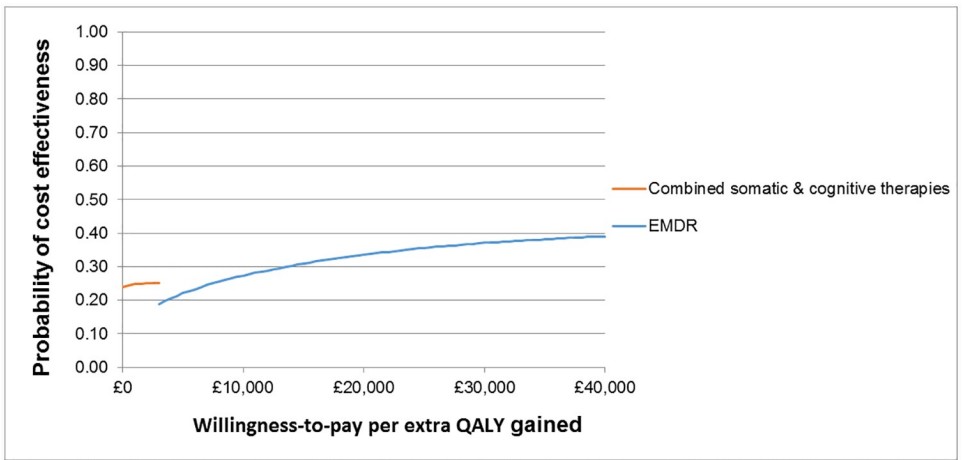

**Fig 3. Cost-effectiveness acceptability frontier: Base-case analysis.**

TF-CBT being reduced. However, results for other interventions were not affected. Results of deterministic sensitivity analyses are shown in S6 File. For information, results of the NICE guideline economic analysis are shown in S7 File.

## Discussion

### Overview of findings

EMDR appears to be the most cost-effective intervention for adults with PTSD more than 3 months after trauma, followed by combined somatic/cognitive therapies, self-help with support, psychoeducation, SSRIs, TF-CBT, self-help without support, non-TF-CBT and combined TF-CBT/SSRIs. Counselling appears to be less cost-effective than no treatment, due to its relatively low clinical effectiveness (the lowest amongst all active treatments assessed in the economic analysis), as shown in the NMAs that informed the economic analysis [3], which was not enough to offset its relatively high intervention cost. The low effectiveness found for counselling can be attributed to counselling's non-directive person-centred approach, which is less likely to help the person overcome avoidance (which is one of the criteria for PTSD), and thus less likely to reduce PTSD symptoms. On the other hand, counselling's effectiveness may have been underestimated to some extent due to researcher allegiance, since in the vast majority of the RCTs on counselling that informed the economic analysis, counselling served as a control treatment to other active interventions, primarily TF-CBT [3].

Results were characterised by uncertainty, as reflected in relatively low probabilities of each intervention being cost-effective amongst alternative treatment options; however they were overall robust to assumptions tested through deterministic sensitivity analyses.

### Strengths and limitations

Our analysis utilised effectiveness data derived from a systematic review and NMA of changes in PTSD symptoms [3]. This methodology enabled us to consider information from direct and indirect comparisons between interventions, and allowed simultaneous comparisons across all options while preserving randomisation [35]. This approach for evidence synthesis is essential for populating model-based economic studies assessing more than two competing interventions. No inconsistency was detected between direct and indirect evidence. We used 10,000 iterations of the NMA models in the economic analysis, which represent the full uncertainty and correlation in the relative effects.

The NMA that informed the base-case economic analysis (changes in PTSD symptoms between baseline and treatment endpoint) used a large evidence base and produced robust data. The NMA of 1-4-month follow-up changes in PTSD symptoms, which informed the secondary analysis, showed considerable uncertainty for most interventions, due to the small number and size of the included studies; TF-CBT and EMDR were the only treatments in this NMA with data on at least 100 people at 1-4-month follow-up that showed evidence of sustained effect. Thus, results of this secondary economic analysis should be interpreted with caution. Both NMAs were characterised by high between-trial heterogeneity, which is likely to have been caused by heterogeneity across populations included in the trials considered in the NMAs, for example, in terms of the presence of a formal PTSD diagnosis, the severity, complexity and chronicity of PTSD symptoms, the type, extent and multiplicity of trauma exposure, the presence of comorbidity, and also the variability of interventions within each assessed option and the differences across settings, e.g. inpatient versus outpatient delivery of interventions [3]. Heterogeneity may also have been caused by the type, multiplicity and timing of previous treatments in trial participants, but relevant information was not available in the

majority of the RCTs included in the NMAs; therefore, the impact of previous treatments on the effectiveness of the interventions cannot be assessed.

Regarding the potential heterogeneity across populations in the trials that informed the economic model, it should be noted that, of the 82 studies that were included in the NMAs that informed the economic analysis, 50 (61%) recruited people with a formal PTSD diagnosis whereas the remaining 32 (39%) recruited people with clinically important PTSD symptoms, as indicated by baseline scores above a predefined threshold on a validated PTSD symptom scale. The percentage of trials recruiting people with a formal PTSD diagnosis was relatively high (range 70–100%) among trials assessing TF-CBT alone or combined with SSRIs, EMDR and counselling; moderate (range 50–66%) among trials assessing non-TF-CBT and self-help with support; and rather low (range 25–33%) among trials assessing combined somatic/cognitive therapies, self-help without support and psychoeducation ([3]; see S5 Appendix for population characteristics of included studies in the NMA). It may be hypothesised that a formal diagnosis of PTSD is associated with more severe symptoms at baseline, resulting in interventions tested on such populations appearing to be less clinically and cost-effective compared with interventions tested on people with clinically important PTSD symptoms, who may have had less severe symptoms at baseline. Nevertheless, the NMA and economic modelling results did not confirm such a hypothesis, since, for example, EMDR and, to a lesser degree, TF-CTB, showed high clinical and cost-effectiveness, despite both having been tested in RCTs that, in their majority, recruited people with a formal PTSD diagnosis. On the other hand, it is possible that participants in the trials that recruited people with clinically important PTSD symptoms might meet criteria for (and might have received) a formal PTSD diagnosis, but they were not required to do so in order to participate in the trial, and this could well have been for pragmatic reasons related to trial management rather than an intention to recruit people with potentially lower symptom severity. Furthermore, for PTSD symptom scales that are based closely on diagnostic criteria, e.g. PTSD Checklist (PCL), scoring above the clinical threshold may be regarded as comparable to receiving a formal diagnosis of PTSD. Related to this point, we should note that all RCTs that evaluated combined somatic/cognitive therapies and recruited people with clinically important PTSD symptoms used scales that are based on diagnostic criteria (e.g. PCL and Modified PTSD Symptom Scale [MPSS-SR]); similarly, the majority of the studies that evaluated psychoeducation and recruited people with clinically important PTSD symptoms used scales based on diagnostic criteria (e.g. PCL and Davidson Trauma Scale). Consequently, the symptom severity of trial populations receiving combined somatic/cognitive therapies and psychoeducation (who, in the majority of trials, were not required to have a formal diagnosis of PTSD) was likely similar to the symptom severity of populations with a formal diagnosis of PTSD receiving other interventions considered in our NMA and economic analysis.

The strengths and limitations of the NMAs that informed the economic analyses should be considered when interpreting the cost-effectiveness results. Moreover, the quality and limitations of RCTs considered in the NMAs have unavoidably impacted on the quality of the model input parameters. Although all interventions included in the economic analysis had been tested on at least 100 trial participants for treatment endpoint, the size of the evidence base differed considerably across interventions. TF-CBT had by far the largest evidence base in both NMAs (N = 903 across 29 RCTs at treatment endpoint and N = 753 across 13 RCTs at 1-4-month follow-up), which gives us more confidence in the results on its clinical (and, consequently, cost-) effectiveness. The evidence base was more limited for other interventions included in the economic analysis (each tested on N<350 in each of the two NMAs). For comparison, EMDR, which was shown to be the most cost-effective treatment option in our economic analysis, was tested on N = 260 across 11 RCTs at treatment endpoint and N = 121 across 4 RCTs at 1-4-month follow-up in the NMAs that informed the economic analysis [3].

The economic model structure did not incorporate discontinuation due to limited data availability. However, the NMAs that informed the economic analysis utilised intention-to-treat data, where available, so that discontinuation has been implicitly considered in the economic analysis. The probabilistic analysis took into account, where possible, the completion rates of the interventions in the RCTs that informed the economic analysis, so that the number of sessions reflected, up to a degree, the attrition rates of each intervention. The time horizon of the analysis was 3 years, in order to capture longer-term effects and costs associated with a course of treatment for PTSD without significant extrapolation over the natural course of PTSD.

The baseline risk of remission was estimated from a large longitudinal study on adults with PTSD in the community [12], as the survey's target population was deemed to be directly relevant to our study population. The risk of relapse was not available in published literature, and was therefore based on expert opinion. Utility data were derived from a systematic literature review. Costs incurred by adults with PTSD and those remitting from PTSD were based on published national survey data, supplemented with other published evidence and expert opinion, due to lack of more accurate information. Sensitivity analysis showed that results were robust to use of alternative values for the risk of relapse, utility and costs. The risk of side effects from SSRIs was based on an uncontrolled study that did not examine the rate of side effects that were attributable to SSRIs. Therefore, our economic analysis may have overestimated the impact of common side effects from SSRIs relative to other treatments and thus may have underestimated the relative cost effectiveness of SSRIs.

In conclusion, our study is characterised by different strengths and limitations, which we have considered when constructing our model and interpreting the results of our analysis. We carried out probabilistic analyses, which took into account the uncertainty around model parameters and, where possible, we conducted secondary and deterministic sensitivity analyses to address uncertainties and gaps in the evidence.

## Comparison with existing economic evidence

Published economic evaluations of interventions for PTSD in adults have concluded that exposure therapy (a form of TF-CBT) is more cost-effective than no treatment [36]. TF-CBT and SSRIs are likely more cost-effective than usual care [18]; prolonged exposure (TF-CBT) has been found to be more cost-effective than SSRIs [37]. Finally, self-management was shown to be no more effective but overall less costly than psychoeducation [38]. These economic studies evaluated a limited range of interventions for adults with PTSD and made very few comparisons between active interventions; notably, EMDR, which was shown to be the most cost-effective intervention in our analysis, has not been evaluated in previously published economic literature on adults with PTSD.

Overall, our findings are in agreement with previously published evidence. Our economic analysis estimated the cost-effectiveness of a wider range of interventions available for adults with PTSD, such as EMDR, combined somatic/cognitive therapies, self-help, non-TF-CBT and counselling and allowed, for the first time, simultaneous comparisons of cost-effectiveness across interventions, and their ranking from the most to the least cost-effective.

On the other hand, an economic evaluation of psychological interventions for PTSD in children and young people, which also used efficacy data derived from a NMA and adopted a similar approach and methodology to the analysis described here, concluded that individual forms of TF-CBT were most cost-effective in the treatment of children and young people with PTSD, whereas EMDR occupied middle cost-effectiveness rankings amongst the treatment options assessed [39]. This finding was attributed to the lower effectiveness of EMDR relative to other treatments in children and young people compared with adult populations [40].

## Generalisability of the results and implications of the study

Our analysis was conducted from the perspective of the NHS/PSS in England. Results may be generalisable to other settings with similar funding and structure of healthcare and personal social services and comparable care pathways for adults with PTSD. Conclusions on cost-effectiveness ultimately rely on the cost-effectiveness threshold adopted, and this depends on the policy makers' willingness-to-pay for treatment benefits, which may vary across countries and health systems.

Our analysis estimated the resource use relating to the delivery of each intervention based on information reported in the RCTs that informed the economic analysis; for example, the mode number of hours for a course of EMDR and TF-CBT was 9 and 13.5, respectively. If the duration and therefore the cost of an intervention is considerably different from our estimates, then its relative cost-effectiveness is expected to be affected. However, reducing the number of sessions of an intervention will improve its cost-effectiveness only if its clinical effectiveness remains unaffected. In practice, a reduction in the number of sessions below a point that is critical for the optimal delivery of the intervention is expected to reduce its clinical effective-ness, too; the impact on its cost-effectiveness will depend on the trade-off between a lower intervention cost and a lower clinical effectiveness.

Based on the results of the NMAs and the economic analysis, the NICE guideline on PTSD recommended EMDR and individual TF-CBT for the treatment of adults with PTSD presenting more than 3 months after trauma [5]. Both interventions were shown to be effective in reducing PTSD symptoms post-treatment and were the only ones with sufficient evidence to suggest sus-tainment of effect beyond treatment. EMDR appeared to be the most cost-effective intervention amongst those assessed. TF-CBT appeared to be less cost-effective than other interventions (i.e. combined somatic/cognitive therapies, psychoeducation, self-help with support and SSRIs), but had by far the largest evidence base and the guideline economic analysis showed that brief indi-vidual TF-CBT (delivered in fewer than 8 sessions) had the highest clinical and cost effectiveness amongst all options assessed; the finding that brief individual TF-CBT had the highest clinical effectiveness was explained by inspection of the clinical data, which revealed that participants in trials of brief individual TF-CBT had less severe PTSD symptoms at baseline, and therefore were likely to have a better response to treatment, compared with participants in trials of more inten-sive forms of individual TF-CBT. The recommendation for EMDR was restricted to people with non-combat-related trauma, as evidence suggested a non-significant effect on people with com-bat-related trauma.

The NICE guideline recommendations on TF-CBT and EMDR for adults with PTSD are consistent with other published PTSD clinical practice guidelines (compared in [41]). Three more guidelines make recommendations of equal strength for TF-CBT and EMDR [42–44], whereas one guideline makes a strong recommendation for TF-CBT while EMDR has been given a moderate rating [45].

Self-help with support was shown to be the third most cost-effective option amongst those assessed, owing to a combination of its high effectiveness at treatment endpoint (informed by N = 198 across 5 RCTs in the respective NMA [3]) and its low intervention cost. There was also limited evidence (N = 85 across 3 RCTs in the respective NMA [3]) that it can sustain effects beyond treatment endpoint. All 5 RCTs on self-help with support that informed the economic analysis focused on computerised TF-CBT, which is consistent with TF-CBT deliv-ered by a therapist, and this element may have been the driver of the intervention's clinical effectiveness. The NICE guideline committee considered the clinical and cost-effectiveness of self-help with support and, also, that of SSRIs, but noted their narrower evidence base and made weaker ('consider') recommendations for people who expressed a preference for these interventions, and, in the case of self-help, did not have severe PTSD symptoms and were not

at risk of harm to themselves or others. Based on its middle rankings in the NMA and economic analysis, a 'consider' recommendation was also made for non-TF-CBT targeted at specific symptoms, for people who are unable or unwilling to engage in a trauma-focused intervention or have residual symptoms after treatment. Psychoeducation was shown to be cost-effective based on limited and inconclusive clinical evidence; therefore, it was not recommended as a stand-alone intervention, but as part of individual TF-CBT. Finally, the committee noted the evidence of high clinical and cost-effectiveness for combined somatic/cognitive therapies, but also considered their particularly limited evidence base beyond treatment endpoint and the lack of specific indications for these interventions, and decided not to recommend them but instead to make a recommendation for further research [5].

## Conclusion

EMDR appears to be the most cost-effective intervention for adults with PTSD more than 3 months after trauma, followed by combined somatic/cognitive therapies, self-help with support, psychoeducation, SSRIs, TF-CBT, self-help without support, non-TF-CBT and combined TF-CBT/SSRIs. Counselling appears to be less cost-effective than no treatment. Results were characterised by uncertainty, and relatively limited evidence base for interventions other than TF-CBT. There is a need for well-conducted studies that examine the relative clinical and cost-effectiveness of a range of psychological treatments for adults with PTSD, in particular assessment of longer-term costs and effects, to reduce the uncertainty and limitations characterising current evidence.

## Supporting information

**S1 File. Selection of effectiveness data and transformation for use in the economic analysis.**
(DOCX)

**S2 File. Estimation of the baseline probability of remission.**
(DOCX)

**S3 File. Estimation of the unit cost of therapists delivering psychological interventions for PTSD in the British National Health Service (NHS).**
(DOCX)

**S4 File. Estimation of annual health and personal social service costs incurred by adults with PTSD and adults without PTSD.**
(DOCX)

**S5 File. Results of secondary probabilistic economic analysis [beneficial effect up to 3-months post-treatment].**
(DOCX)

**S6 File. Results of deterministic sensitivity analyses.**
(DOCX)

**S7 File. Results of the NICE guideline economic analysis.**
(DOCX)

**S1 Appendix. Search strategy.**
(DOCX)

**S2 Appendix. Study protocol.**
(DOCX)

**S3 Appendix. Details of the statistical analysis and WinBUGS codes for data synthesis.**
(DOCX)

**S4 Appendix. Details of the inconsistency checks and WinBUGS codes for inconsistency models.**
(DOCX)

**S5 Appendix. Characteristics of studies included in the network meta-analysis, and full references.**
(DOCX)

**S6 Appendix. List of excluded studies with reasons for exclusion.**
(DOCX)

**S7 Appendix. NMA data files.**
(DOCX)

**S8 Appendix. Risk of bias of studies included in the NMA.**
(DOCX)

**S9 Appendix. Model fit statistics.**
(DOCX)

**S10 Appendix. Inconsistency checks.**
(DOCX)

**S11 Appendix. Relative effects between all pairs of interventions: Direct, indirect and combined (NMA) results.**
(DOCX)

**S12 Appendix. Results of the NICE guideline NMA.**
(DOCX)

**S13 Appendix. Pairwise sub-analyses.**
(DOCX)

**S14 Appendix. References in the online supplementary material.**
(DOCX)

## Acknowledgments

We thank other members of the Guideline Committee for the NICE guideline on 'Post-traumatic stress disorder' for their contributions to this work. Members of the Guideline Committee were: Steve Hajioff, Philip Bell, Gita Bhutani, Sharif El-Leithy, Neil Greenberg, Nick Grey, Cornelius Katona, Jonathan Leach, Richard Meiser-Stedman, Rebecca Regler, Vikki Touzel, and David Trickey.

## Author Contributions

**Conceptualization:** Ifigeneia Mavranezouli, Odette Megnin-Viggars, Nick Grey, Gita Bhutani, Jonathan Leach, Caitlin Daly, Sofia Dias, Nicky J. Welton, Cornelius Katona, Sharif El-Leithy, Neil Greenberg, Sarah Stockton, Stephen Pilling.

**Data curation:** Ifigeneia Mavranezouli, Odette Megnin-Viggars, Nick Grey, Gita Bhutani, Jonathan Leach, Caitlin Daly, Sofia Dias, Sarah Stockton.

**Formal analysis:** Ifigeneia Mavranezouli.

**Funding acquisition:** Stephen Pilling.

**Methodology:** Ifigeneia Mavranezouli, Odette Megnin-Viggars, Caitlin Daly, Sofia Dias, Nicky J. Welton.

**Project administration:** Ifigeneia Mavranezouli.

**Validation:** Nick Grey, Gita Bhutani, Jonathan Leach, Cornelius Katona, Sharif El-Leithy, Neil Greenberg, Stephen Pilling.

**Visualization:** Ifigeneia Mavranezouli.

**Writing – original draft:** Ifigeneia Mavranezouli.

**Writing – review & editing:** Odette Megnin-Viggars, Nick Grey, Gita Bhutani, Jonathan Leach, Caitlin Daly, Sofia Dias, Nicky J. Welton, Cornelius Katona, Sharif El-Leithy, Neil Greenberg, Sarah Stockton, Stephen Pilling.

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
