## [Decision Letter · Decision Letter 0]

25 Nov 2019

PONE-D-19-20536

Cost-effectiveness of psychological treatments for post-traumatic stress disorder in adults

PLOS ONE

Dear Dr Mavranezouli,

Thank you for submitting your manuscript to PLOS ONE. After careful consideration, we feel that it has merit but does not fully meet PLOS ONE’s publication criteria as it currently stands. Therefore, we invite you to submit a revised version of the manuscript that addresses the points raised during the review process.

We would appreciate receiving your revised manuscript by Jan 09 2020 11:59PM. To enhance the reproducibility of your results, we recommend that if applicable you deposit your laboratory protocols in protocols.io, where a protocol can be assigned its own identifier (DOI) such that it can be cited independently in the future. For instructions see: http://journals.plos.org/plosone/s/submission-guidelines#loc-laboratory-protocols

We look forward to receiving your revised manuscript.

Kind regards,

Scott McDonald

Academic Editor

PLOS ONE

Journal Requirements:

I have read the journal's policy and the authors of this manuscript have the following competing interests: SD and NJW were co-applicants on a grant (unrelated to this work) from the MRC Methodology Research Programme which included an MRC Industry Collaboration Agreement with Pfizer Ltd, who part-funded a researcher to work on statistical methodology. GB is a co-investigator on a NIHR RfPB grant, Eye Movement Desensitization and Reprocessing Therapy in Early Psychosis (EYES): A feasibility randomised controlled trial. NGreenberg is the Royal College of Psychiatrists Lead for Military and Veterans’ Health and is a trustee of two military charities. He is also a senior researcher with King’s College London working on a number of military mental health studies. NGrey is a member of the Wellcome Trust Anxiety Disorders Group developing, testing and disseminating Cognitive Therapy for PTSD (CT-PTSD), a trauma-focused cognitive behavioural therapy (TF-CBT). He has published papers and book chapters on CT-PTSD, and facilitates teaching workshops for which payment is received. As editor, he receives royalties from sales of a trauma book, A Casebook of Cognitive Therapy for Traumatic Stress Reactions. CK is Medical Director of the Helen Bamber Foundation (a human rights charity) and refugee and asylum mental health lead for the Royal College of Psychiatrists. He writes expert psychiatric reports in the context of asylum mental health. JL is NHS England Medical Director for Military and Veterans Health. SP receives funding from NICE for the development of clinical guidelines and is also supported by the NIHR UCLH Biomedical Research Centre. The authors report no other relationships or activities that could appear to have influenced the submitted work.

We note that you received funding from a commercial source: Pfizer Ltd,

Reviewers' comments:

Reviewer's Responses to Questions

**Comments to the Author**

1. Is the manuscript technically sound, and do the data support the conclusions?

Reviewer #1: Partly

Reviewer #2: Yes

Reviewer #3: Yes

2. Has the statistical analysis been performed appropriately and rigorously? 

Reviewer #1: I Don't Know

Reviewer #2: Yes

Reviewer #3: Yes

3. Have the authors made all data underlying the findings in their manuscript fully available?

Reviewer #1: No

Reviewer #2: Yes

Reviewer #3: Yes

4. Is the manuscript presented in an intelligible fashion and written in standard English?

Reviewer #1: Yes

Reviewer #2: Yes

Reviewer #3: Yes

5. Review Comments to the Author

Reviewer #1: This archival data study addresses an important and under-studied question, and reports that EMDR was most cost-effective although TF-CBT had the strongest evidence base amongst 11 psychotherapy and pharmacotherapy approaches for PTSD treatment in the UK National Health Service.

Overall, the paper read more like a technical report than a journal article. As a non-economist, I found it very difficult to follow and disentangle all of the detailed assumptions and metrics required to calculate cost-effectiveness. Several specific limitations also made it difficult to accept the findings as meaningful for actually prioritizing treatments:

1. The treatments were proposed to be delivered on a short-term (3-month, with 3-months follow-up for the SSRI) basis. This may be the official norm in the NHS but it is not reflective of the actual clinical picture in treating chronic PTSD, and therefore is not applicable except when PTSD is relatively acute and resolves rapidly.

2. The low overall probability of remission in the 3-month treatment period and the 1-4 month follow-up period and frequent relapse (for which the .10 probability over 3 months seems very low, even if some “experts” opined that to be the case) means that for large majorities of patients receiving any of these treatments there will need to be continuing or alternative treatment and increasing costs (including due to impairment as well as further treatment, and related healthcare and services). Thus, the cost-effectiveness results are at best potentially relevant to very short-term outcomes, and at worst greatly over-estimate the value of the treatments. Over longer periods of time, it is likely that the most inexpensive and readily available help, self-help with or without support, will be most cost-effective but it is of limited effectiveness with PTSD that is more than mild in severity.

3. The data on effectiveness drawn from the in-review network meta-analysis are probably based on randomized clinical trials or evaluation studies that do not reflect the actual effects when these treatments are delivered in standard care.

4. The inclusion of patients with no diagnosis but above threshold self-reported symptoms means that many will have at most mild PTSD and recovery will be much more likely than with full (let alone chronic) PTSD. This further weights the results toward less clinically complex cases, which are the types of cases often tested in studies of EMDR and TF-CBT, so this may bias the results in favor of those interventions.

5. The differences in the overall outcome, NMB, seem relatively small (approximately 500-1000 pounds per person) for the top four interventions, so some justification of why they are meaningful differences is needed.

6. The cost-effectiveness differences are all essentially due to costs, because the QALY estimates are almost identical except for counselling. This suggests that small changes in the number of sessions provided fort the professional-delivered interventions will greatly alter cost-effectiveness (as is tangentially noted in the Discussion with the mention of the increased cost-effectiveness of TF-CBT if delivered for 8 sessions). Thus, rather than demonstrating the superior cost-effectiveness of any intervention this analysis seems mainly to show that the more extensive the professional time/costs involved the less cost-effective are the formal interventions, which is not surprising.

7. The comment about TF-CBT’s evidence base is not relevant because that is not the focus of the analyses presented.

Even with these limitations acknowledged, and the Abstract and Discussion revised to make the conclusions more cautious in endorsing EMDR and more inclusive in noting the cost-effectiveness of self-help with and without social support (despite its low effectiveness), the paper seems better suited for a journal audience of policymakers, administrators, or behavioral economists than a broader health professional/researcher audience.

Reviewer #2: The manuscript, “Cost-effectiveness of Psychological Treatments for Post-traumatic Stress Disorder in Adults,” covers an important topic with broad systemic and individual-level implications. Moreover, this article highlights the need to find a balance between cost-effectiveness and treating patients with clinically-indicated treatments with a solid evidence base. Finally, the use of a the quality of life outcome adds to the richness of the analysis beyond symptom reduction.

Introduction/Methods:

This manuscript would benefit from greater clarification regarding the treatments examined. While many of my questions were answered within the supplemental “PTSD adult NMA manuscript,” readers would benefit from having these treatments explained. Even after reviewing the supplemental manuscript, I am unsure which treatments qualify as “counseling” and what, if anything, differentiates the practitioners of counseling from the other treatments. It is unclear to me to what extent settings differ across these treatment types and how that affects the outcomes. For example, is it the restrictiveness/intensity of the treatment setting, as in inpatient treatment, that drives the cost-effectiveness of the treatment, the level of training from the provider, or researcher allegiance effects, more so than the type of treatment?

1) What qualifies as counseling (Interventions Section, p. 11)?

2) Which therapies are considered “combined somatic/cognitive therapies”? Is this classification determined by the length of sessions and level of therapist training (4 individual sessions by a band 7 therapist) or rather eclectic treatments which would not qualify as CBT, for example? (Interventions Section, p. 11)

3) What is the role of the treatment setting on the outcomes? For example, you mention in the Interventions section (p.11), that TF-CBT was divided by number of sessions and format of delivery in the economic analysis informing the NICE guideline, but that in the analyses here it was considered one treatment option. How might the results differ based on this grouping?

Results:

4) On Table 3 (p. 24), the probability of cost effectiveness of TF-CBT is listed as .26. The manuscripts notes elsewhere that the mode number of sessions was 13.5, yet when it is delivered in fewer than eight sessions it becomes the most cost-effective treatment. It seems that number of sessions becomes a critical factor in determining the cost effectiveness. It may be helpful to have a discussion or a range of results to reflect changes by number of sessions or other important factors.

5) Counseling (p. 24) is listed as less cost effective than no treatment. Which factors do the authors believe contribute to this finding?

6) A reference would be helpful for readers to examine the evidence base for EMDR given it’s controversial nature (e.g., Shaprio & Brown, 2019; American Psychological Association, Guideline Development Panel for the Treatment of PTSD in Adults, 2017).

Conclusions/Limitations

7) The manuscripts notes (p. 30) that TF-CBT appeared to be less cost-effective than other interventions, yet had “by far the most solid evidence base.” What implications does that have on the findings? Perhaps TF-CBT yields stronger maintenance of treatment gains over time; although it appears that comparable remission rates were unavailable for most other treatments.

8) In addition to grouping together TF-CBT by number of sessions, the limitations section of the supplemental manuscript notes that “TF-CBT” includes a broad range of therapies from CPT to mindfulness-based cognitive therapy, whereas other studies (e.g., Gerger et al., 2014) assess CBT, CT, and ET both separately and as a group. Given the high variation among interventions and between-trial heterogeneity, what might be the impact this grouping?

9) How might the timeline and complexity of the trauma histories impact the findings?

10) How do the included studies compare in terms of patient’s treatment histories (e.g., history of multiple psychological treatments and the timing of these treatments)?

Overall, this was a well-written manuscript representing an important contribution to PTSD research, while highlighting important gaps in the literature (e.g., long-term follow-ups, applicability of treatments to complex traumas).

Reviewer #3: This paper presents results from a detailed and extensive programme of work. It draws on a separately reported strand of analysis on clinical effectiveness of a range of different types of interventions for adults with PTSD. It builds on that work by combining it with new estimations of the costs of each type of intervention when delivered in England, so that both clinical-effectiveness and cost (both to services and to individuals) are consdiered . The paper presents the methods used to inform the NICE guidelines for PTSD treatment in England: in England NICE guidelines are absolutely pivotal to informing NHS clinical practice and resourcing decisions.

While much of this evidence may exist (in different form) in the NICE documentation alongside the guidelines, it is so important that the research underpinning guidelines is also made available in peer review journal form – for transparency and so it is indexed and identifiable for systematic review. The analysis in this paper also used some different parameters to inclusion to the guidelines. It would be useful if the authors could comment on what impact on results increasing the minimum number of participants in included studies from 50 to 100 had on the overall results (I might have missed this).

Please note I do not have great expertise in clinical cost-effectiveness calculations and do not feel able to properly comment on those methods. The process has been overseen by expert committee (some of whom are also authors), appears detailed, and is well-documented. The figures aid interpretation of methods as well as of findings, esp fig 1.

Such costings inevitably must deal in averages. The decisions on where to anchor those averages seems sensible (although the community data finds rates of screen-positive PTSD to be highest in 16-24 year olds, while the cohort anchors initiation age at 39 to reflect the treatment population. This may well be appropriate.

However, in the limitations it would be good to acknowledge that while – overall – EMDR may be the most cost-effective, this could vary with group characteristics. E.g. some interventions may be more clinically effective (and therefore, potentially, also more cost-effective) in the youngest age group, or in those with the most (or least) severe symptoms, or those with comorbid or delayed PTSD. Just something to acknowledge. An understanding of this is important in understanding the applications of the results, and how rigidly the recommendations should be applied in practice. There may be situations where a lower ranked intervention is the most appropriate.

I would have liked a line defining each intervention, or at least describing how the interventions evaluated were assigned to the intervention classifications. For example, ‘counselling’ gets a damning verdict – but I am not clear precisely what kinds of interventions were counted here. Are they simply interventions that were so poorly done that they were unclassifiable in any of the other categories? Which could be a confounding factor explaining their poor performance? Were all interventions assigned to one category only, or was there overlap?

Abstract has a sentence which reads: ‘TF-CBT has the most solid evidence base’ – take especial care in the abstract (which will be all that many read) that it is clear what this means (the usual disentangling ‘lack of evidence’ and ‘evidence of lack’ is needed).

6. PLOS authors have the option to publish the peer review history of their article (what does this mean?). If published, this will include your full peer review and any attached files.

Reviewer #1: No

Reviewer #2: Yes: Sarah M. Scott, Ph.D

Reviewer #3: No

---

## [Author Response · Author response to Decision Letter 0]

8 Jan 2020

Responses to reviewers' comments:

We thank the reviewers for their comments. Please note that the manuscript pages in our responses refer to the version of the revised manuscript submitted with tracked changes.

Reviewer #1:

This archival data study addresses an important and under-studied question, and reports that EMDR was most cost-effective although TF-CBT had the strongest evidence base amongst 11 psychotherapy and pharmacotherapy approaches for PTSD treatment in the UK National Health Service.

Overall, the paper read more like a technical report than a journal article. As a non-economist, I found it very difficult to follow and disentangle all of the detailed assumptions and metrics required to calculate cost-effectiveness.

RESPONSE: we thank the reviewer for their comment. We gave details on all assumptions and metrics required for the economic analysis to ensure transparency. We provided the standard level of detail used to support a model-based economic evaluation paper, as recommended in the literature. See for example:

Husereau et al. Consolidated Health Economic Evaluation Reporting Standards (CHEERS) statement. Cost Effectiveness and Resource Allocation 2013; 11(6)

https://doi.org/10.1186/1478-7547-11-6

And 

Husereau et al. Consolidated Health Economic Evaluation Reporting Standards (CHEERS)—Explanation and Elaboration: A Report of the ISPOR Health Economic Evaluation Publication Guidelines Good Reporting Practices Task Force. Value in Health 2013; 16(2): 231-50.

https://doi.org/10.1016/j.jval.2013.02.002

Several specific limitations also made it difficult to accept the findings as meaningful for actually prioritizing treatments:

1. The treatments were proposed to be delivered on a short-term (3-month, with 3-months follow-up for the SSRI) basis. This may be the official norm in the NHS but it is not reflective of the actual clinical picture in treating chronic PTSD, and therefore is not applicable except when PTSD is relatively acute and resolves rapidly.

RESPONSE: The economic analysis has assessed the cost-effectiveness of first-line interventions (provided, on average, over 3 months), but treatment for people with PTSD is not limited to 3-months in the model. People not remitting following first-line treatment or relapsing (i.e. all people in the PTSD health state) were assumed to receive further treatment comprising standard care, as reflected in the PTSD state costs (see page 16, lines 19-28 and S4 file, which provides annual health and personal social service use and costs incurred by people with PTSD and those without PTSD). Resource use of standard care was estimated based on a national psychiatric morbidity survey conducted in England in 2014, supplemented with resource use data from other published sources and expert opinion, which were subsequently combined with national unit costs, as reported on page 16. This longer-term further treatment was common to all arms of the economic model and included primary, community and secondary healthcare as well as personal social services. We have now added text under ‘Model structure’ to clarify this (page 8, lines 11-13). 

Please note that these costs were estimated over a 3-year period, which was the time horizon of the analysis. As stated on page 8, lines 17-19, this time frame was deemed adequate to capture longer-term costs and effects of treatment, without making significant extrapolations and assumptions over the course of PTSD.

2. The low overall probability of remission in the 3-month treatment period and the 1-4 month follow-up period and frequent relapse (for which the .10 probability over 3 months seems very low, even if some “experts” opined that to be the case) means that for large majorities of patients receiving any of these treatments there will need to be continuing or alternative treatment and increasing costs (including due to impairment as well as further treatment, and related healthcare and services). Thus, the cost-effectiveness results are at best potentially relevant to very short-term outcomes, and at worst greatly over-estimate the value of the treatments. Over longer periods of time, it is likely that the most inexpensive and readily available help, self-help with or without support, will be most cost-effective but it is of limited effectiveness with PTSD that is more than mild in severity.

RESPONSE:

The probability of remission in the 3-month treatment period was estimated to be 0.044 for no treatment (which was used as baseline), based on data from a national survey, as explained in S2 file. The probability of remission in the 3-month treatment period for each of the active interventions was estimated by applying the relative effects (expressed as odds ratios) of each intervention versus no treatment, obtained from the network meta-analysis (and reported in Table 2), onto the baseline probability of remission for no treatment. This gave 3-month probabilities of remission ranging from 0.15 (counselling) to 0.65 (EMDR). Following that period, and for the remaining duration of the model, we conservatively applied probabilities of remission using the survey data across all arms of the model.

People who remitted were at risk of relapse for the whole duration of the economic model. The risk of relapse was based on advice and consensus amongst experts with clinical experience in treating people with PTSD, as we were not able to identify reliable long-term relapse data from relevant longitudinal studies. This is standard recommended source of information in order to populate economic models when relevant published literature is not available. We would be grateful if the reviewer could direct us to a reliable source of relapse rates for people with PTSD. Nevertheless, we did sensitivity analysis were the relapse risk of 0.10 was varied between 0.05 and 0.20; results of this analyses, which are reported in S6 file, show that the findings of the base-case analysis were not affected, with the exception of TF-CBT which dropped one place in the cost-effectiveness ranking. This finding has already been reported in the manuscript (page 28, lines 140-143).

As we explain in response to another comment by the reviewer, people not remitting after receiving one of the assessed interventions and people relapsing following remission (i.e. all people in the PTSD state) were assumed to receive standard care for PTSD and to incur primary, community and secondary healthcare as well as personal social service costs for the remaining duration of the economic model, the time horizon of which was 3 years. 

Therefore, our cost-effectiveness results are not only relevant to very short-term outcomes, but cover a period of 3 years. We also disagree with the view that over longer periods of time it is likely that inexpensive and readily available self-help, with or without support, will be most cost-effective. Our analysis showed that self-help without support is not cost-effective over 3 years, although it was the most inexpensive intervention (regarding intervention cost); self-help with support was amongst the most cost-effective options, but this finding was partly driven by the effectiveness of self-help, which ranked 4th best intervention in terms of effectiveness, as it can be seen in Table 2. We also note that, as can be seen in Table 3, EMDR was the most cost-effective intervention although it had one of the highest intervention costs. So, it is not the cost alone that determines the cost-effectiveness of an intervention, it is the combination of its cost and effectiveness, in relation to respective combinations of cost and effectiveness for other alternative treatment options.

3. The data on effectiveness drawn from the in-review network meta-analysis are probably based on randomized clinical trials or evaluation studies that do not reflect the actual effects when these treatments are delivered in standard care.

RESPONSE: The data on relative effects were obtained from a network meta-analysis of RCTs, as the RCT design is the gold standard for estimating relative effects. However, relative effects were applied onto baseline (no treatment) absolute remission rates of PTSD derived from a national survey in the general population, to obtain estimates of treatments’ absolute effects under standard care delivery.

4. The inclusion of patients with no diagnosis but above threshold self-reported symptoms means that many will have at most mild PTSD and recovery will be much more likely than with full (let alone chronic) PTSD. This further weights the results toward less clinically complex cases, which are the types of cases often tested in studies of EMDR and TF-CBT, so this may bias the results in favor of those interventions.

RESPONSE: We do not agree with the view that patients with no formal PTSD diagnosis but who are above the threshold on a PTSD scale (which may be self-reported but may also be clinician-rated) will have necessarily mild PTSD. Also, we do not have reasons to believe a-priori that EMDR and TF-CBT have been tested on less clinically complex cases. In fact, the evidence dataset that was used to inform the economic model includes the following populations:

As we report in our NMA paper (accepted for publication in Psychological Medicine), of the 90 studies that were included in our NMAs, 58 (64%) included patients with a formal PTSD diagnosis.

Of the 84 studies that reported continuous data and were included in the NMAs that informed the economic analysis, 52 (62%) included patients with a formal PTSD diagnosis.

Regarding studies testing each intervention considered in the economic analysis, the percentage of studies with a formal PTSD diagnosis was:

• Trauma-focused CBT: 29/37 (78.3%)

• Trauma-focused CBT with SSRIs 3/3 (100%)

• EMDR 9/12 diagnosis (75%)

• Non-trauma-focused CBT 4/8 (50%)

• Combined somatic/cognitive therapies 1/4 (25%)

• Self-help with support 4/6 diagnosis (66%)

• Self-help without support: 3/11 (27%)

• Counselling: 8/11 (73%)

• Psychoeducation: 1/3 (33%)

This information is provided in Appendix 5 of our NMA paper. Note that the number of studies on this list is higher than the total number of studies (84) considered in the economic analysis because some studies assessed more than one active intervention and have been counted twice or even three times.

Studies on EMDR had one of the highest percentages of formal PTSD diagnosis (in contrast to what the reviewer had hypothesised), which, according to the reviewer’s argument, should have biased the results against EMDR, as the populations were more likely to have more complex PTSD. However, EMDR was shown to have the highest efficacy amongst treatments. On the other extreme, studies on self-help without support had one of the lowest percentages of formal PTSD diagnosis, which, according to the reviewer, would mean that the intervention was tested on milder PTSD cases and thus higher effectiveness would be expected, yet it was one of the least effective interventions in the dataset, as shown in Table 2 of the manuscript.

5. The differences in the overall outcome, NMB, seem relatively small (approximately 500-1000 pounds per person) for the top four interventions, so some justification of why they are meaningful differences is needed.

RESPONSE: A higher NMB suggests that an intervention is more cost-effective than its comparator, regardless of the magnitude of the difference between NMBs, although a difference in cost of £500-£1,000 per person over 3 years is not negligible if projected to the total population of people with PTSD receiving care. An intervention with a lower NMB is by definition less cost-effective, and its use instead of the more cost-effective option would result in inefficient use of resources. It needs to be noted that the uncertainty around cost-effectiveness, as reflected in the mean rank and in the probability of the best intervention being cost-effective, should also be taken into account when interpreting the difference in NMBs (and it may be more important than the magnitude of the absolute difference in NMBs when interpreting results). The NMB combines costs and effects (QALYs) into a single metric, to simplify judgements on cost-effectiveness. Please note that NMB represents net monetary benefit, i.e. the intervention’s benefit expressed in pounds (£) minus the intervention’s cost, hence the higher NMB an intervention has, the more cost-effective it is.

Regarding cost differences between 2 interventions, and whether they are meaningful or not: a difference in cost between two interventions of £500-£1000 per person may be considered high if the extra benefit is minimal; it is a waste of money if both interventions have the same effect; and it is a harmful expenditure if the costlier intervention has a worse outcome than its comparator.

Nevertheless, cost-effectiveness results need to be considered alongside uncertainty (as already stated), the magnitude and quality of the evidence base, the (un)suitability of some interventions for specific sub-populations, and other relevant factors; for this reason, cost-effectiveness was not the sole factor that the NICE guideline committee took into account when making recommendations, as reported on pages 33, line 284 to 35, line 323.

6. The cost-effectiveness differences are all essentially due to costs, because the QALY estimates are almost identical except for counselling. This suggests that small changes in the number of sessions provided fort the professional-delivered interventions will greatly alter cost-effectiveness (as is tangentially noted in the Discussion with the mention of the increased cost-effectiveness of TF-CBT if delivered for 8 sessions). Thus, rather than demonstrating the superior cost-effectiveness of any intervention this analysis seems mainly to show that the more extensive the professional time/costs involved the less cost-effective are the formal interventions, which is not surprising.

RESPONSE: we disagree with the view that QALY estimates are almost identical. For example, the difference between EMDR (most effective intervention, 1.80 QALYs) and psychoeducation (1.74 QALYs) was 0.06 QALYs per person over a 3-year period, or 2 QALYs (i.e. 2 years in perfect health) per 100 people in one year. We note that the difference in utility between the state of PTSD and the state of no PTSD in men is 0.09 (according to Table 2), which translates into a difference of 9 QALYs per 100 men moving from the PTSD into the no PTSD state in one year. If EMDR is provided instead of psychoeducation, this leads to 2 additional QALYs, which translates into 2 / 9 = 0.22 or 22 additional men out of 100 moving from the PTSD state into the no PTSD state, i.e. remitting, without relapse, for a year, which is not negligible in our view. It is true that in the area of PTSD treatment there are no great differences between interventions in terms of QALYs, because interventions have an impact primarily on people’s quality of life and have a very limited impact on survival.

The finding that brief individual TF-CBT (delivered in fewer than 8 sessions) was the most cost-effective intervention amongst all options assessed in the guideline analysis was not driven exclusively by the intervention’s cost; it was primarily driven by the fact that the intervention was shown to be the most effective in the respective NMA that informed the guideline analysis, and therefore produced the highest number of QALYs, as shown in S7 file. We have now added this clarification in the manuscript (page 34, lines 294-298).

We disagree with the reviewer’s conclusion that “the more extensive the professional time/costs involved the less cost-effective are the formal interventions”, because this is not what our analysis shows: EMDR costs £400 more than combined somatic/cognitive therapies, £500 more than self-help with support, £600 more than a course of SSRIs and £650 more than self-help without support; yet, it was shown to be more cost-effective than all these options. TF-CBT costs £300 more than non-TF-CBT and is still more cost-effective. On the other hand, self-help without support was by far the least costly intervention and it ranked 7th out of the 11 options assessed. This finding suggests that the conclusion that “the more extensive the professional time/costs involved the less cost-effective interventions are” does not hold. It is the combination of intervention cost and effectiveness of each of the interventions included in the assessment that determine their relative cost effectiveness.

7. The comment about TF-CBT’s evidence base is not relevant because that is not the focus of the analyses presented.

RESPONSE: we feel that our comment is very relevant because a large evidence base (please note that we replaced ‘most solid’ with ‘largest’ throughout the manuscript for clarity) determined the clinical effectiveness of TF-CBT, and a larger evidence base means less uncertainty and more confidence in the findings. It is reassuring for us (and hopefully for the readers and users of the NICE guideline recommendations) that the clinical effectiveness of TF-CBT at treatment endpoint (and, subsequently, its cost-effectiveness) was determined by 29 RCTs, which tested TF-CBT on 903 participants; we would be less confident if the effect of TF-CBT was determined based on e.g. 1 RCT that tested TF-CBT on 10 participants. 

Even with these limitations acknowledged, and the Abstract and Discussion revised to make the conclusions more cautious in endorsing EMDR and more inclusive in noting the cost-effectiveness of self-help with and without social support (despite its low effectiveness), the paper seems better suited for a journal audience of policymakers, administrators, or behavioral economists than a broader health professional/researcher audience.

RESPONSE: limitations of the study have been reported on pages 30-31; we feel that the abstract and discussion report clearly the results and the underlying uncertainty, and also the limitations of the analysis and propose further research to overcome these limitations. The abstract reports the ranking of all interventions, where it is shown that self-help with support was the 3rd most cost-effective intervention while self-help without support was the 7th most cost-effective intervention. We note that self-help with support was also the 4th most clinically effective intervention (as seen in Table 2), which contributed to its high cost-effectiveness; in contrast, self-help without support was one of the least effective interventions, which apparently contributed to its low cost-effectiveness, despite its low intervention costs. We would like to clarify here that with and without support does not mean social support, but clinical support by a therapist; we have now clarified this on page 7, lines 1-2.

Whether the paper is suited for PLOS One is for the editor to judge, but in our experience, economic evaluations of healthcare interventions are widely published in medical and health service research journals and are of interest not only to policy makers and administrators, but also to health professionals who need to make choices guided by efficient use of resources in their everyday practice; they are also of interest to researchers in the area of healthcare who can be guided by the reported methods and gaps in knowledge when conducting their own research. PLOS ONE is a scientific journal that publishes primary research from any discipline within science and medicine, so we feel that our clinical and economic research methods and findings are suited for publication in the journal.

Reviewer #2: 

The manuscript, “Cost-effectiveness of Psychological Treatments for Post-traumatic Stress Disorder in Adults,” covers an important topic with broad systemic and individual-level implications. Moreover, this article highlights the need to find a balance between cost-effectiveness and treating patients with clinically-indicated treatments with a solid evidence base. Finally, the use of a quality of life outcome adds to the richness of the analysis beyond symptom reduction.

RESPONSE: we thank the reviewer for their comment.

Introduction/Methods:

This manuscript would benefit from greater clarification regarding the treatments examined. While many of my questions were answered within the supplemental “PTSD adult NMA manuscript,” readers would benefit from having these treatments explained. Even after reviewing the supplemental manuscript, I am unsure which treatments qualify as “counseling” and what, if anything, differentiates the practitioners of counseling from the other treatments.

RESPONSE: we thank the reviewer for their comment. We have provided definitions of interventions in the related PTSD adult NMA manuscript, but have now also defined the interventions considered in the economic analysis in this manuscript (pages 5-6). Therapists in counselling were not different in the RCTs included in the NMAs that informed the economic analysis. This is also reflected in our estimation of intervention costs, where “all psychological interventions, with the exception of self-help and psychoeducation, were assumed to be delivered in a primary care setting by Band 7 psychological therapists […], to reflect routine practice in the UK” (page 11 line 26 to page 12 line 3).

It is unclear to me to what extent settings differ across these treatment types and how that affects the outcomes. For example, is it the restrictiveness/intensity of the treatment setting, as in inpatient treatment, that drives the cost-effectiveness of the treatment, the level of training from the provider, or researcher allegiance effects, more so than the type of treatment?

RESPONSE: for the economic analysis, all treatments were assumed to be provided in primary care, and this was reflected in the costing of interventions. We have added text to clarify this point (page 11, line 27). All psychological interventions, with the exception of self-help and psychoeducation, were assumed to be delivered by Band 7 psychological therapists, to reflect routine practice in the UK; no difference in training costs were assumed across interventions delivered by Band 7 therapists. Psychoeducation was assumed to be delivered by AfC band 5 Psychological Well-being Practitioners (PWPs) and self-help by AfC band 6 psychological therapists (page 12, lines 3-4). So, the setting did not affect the cost ‘element’ of cost-effectiveness. It needs to be clarified that people not responding to treatment and those relapsing were assumed to receive additional standard care, comprising community, primary and secondary care (and incur relevant costs), which, however, was not determined by the type of intervention received at the start of the model. 

Regarding the setting across RCTs included in the NMAs that informed in the economic analysis, in some RCTs, interventions might have been delivered in an inpatient setting, and this may have contributed to the heterogeneity between trials (this has now been discussed on page 30, lines 188-194). However, there was no systematic difference in the setting (inpatient or outpatient) across interventions considered in the RCTs. 

Some researcher allegiance effects may have been responsible for the low effectiveness of counselling, as this served as the control intervention in a number of studies assessing other active interventions. However, we argue that it is the mechanism of counselling that was primarily responsible for its effect. This has now been discussed on page 29, lines 155-165.

Overall, the difference in cost effectiveness is attributed to the difference in the type of treatment (rather than confounding factors), which translates into different clinical effectiveness and intervention cost.

1) What qualifies as counseling (Interventions Section, p. 11)?

RESPONSE: we have now given the definition of counselling in the paper (page 6, lines 14-22).

2) Which therapies are considered “combined somatic/cognitive therapies”? Is this classification determined by the length of sessions and level of therapist training (4 individual sessions by a band 7 therapist) or rather eclectic treatments which would not qualify as CBT, for example? (Interventions Section, p. 11)

RESPONSE: we have now provided a definition for these interventions on page 6, lines 7-9. As it can be seen, this classification is determined by the content of the interventions and not by delivery characteristics (such as the length of sessions and level of therapist training). 

3) What is the role of the treatment setting on the outcomes? For example, you mention in the Interventions section (p.11), that TF-CBT was divided by number of sessions and format of delivery in the economic analysis informing the NICE guideline, but that in the analyses here it was considered one treatment option. How might the results differ based on this grouping?

RESPONSE: the treatment setting had no impact on the outcomes. However, creating different TF-CBT categories by number of sessions and format had an impact on results because each category had its own clinical effectiveness and intervention cost, as now explained on page 5, lines 23-25. Brief individual TF-CBT was shown to be the most clinically and cost-effective treatment option in the guideline analysis, as reported on page 34, but this result (highest clinical effectiveness) was due to the populations recruited in these trials. We have reported this information on page 34, lines 293-298. Full results of the guideline economic analysis are provided in S7 file.

Results:

4) On Table 3 (p. 24), the probability of cost effectiveness of TF-CBT is listed as .26. The manuscripts notes elsewhere that the mode number of sessions was 13.5, yet when it is delivered in fewer than eight sessions it becomes the most cost-effective treatment. It seems that number of sessions becomes a critical factor in determining the cost effectiveness. It may be helpful to have a discussion or a range of results to reflect changes by number of sessions or other important factors.

RESPONSE: Brief individual TF-CBT was shown to be the most cost-effective option in the guideline analysis not only because it was the lowest number of sessions, but also because it was shown to be the most clinically effective intervention in the guideline NMA. This finding was explained by inspection of the clinical data, which revealed that participants in trials of brief individual TF-CBT had less severe PTSD symptoms at baseline, and therefore were likely to have a better response to treatment, compared with participants in trials of more intensive forms of individual TF-CBT. We have now added this information in the manuscript (page 34, lines 293-298).

Reducing the number of sessions may improve the cost-effectiveness of an intervention if its effectiveness remains unaffected. However, reducing the number of sessions below a ‘critical’ point will also likely reduce its clinical effectiveness, and therefore its cost-effectiveness will be determined by the trade-off between a lower intervention cost and a lower clinical effectiveness. This point has been added on page 33, lines 277-282.

5) Counseling (p. 24) is listed as less cost effective than no treatment. Which factors do the authors believe contribute to this finding?

RESPONSE: a brief explanation for this finding has now been added under results on page 27, lines 114-116 and in more detail under discussion on page 29, lines 155-165.

6) A reference would be helpful for readers to examine the evidence base for EMDR given it’s controversial nature (e.g., Shaprio & Brown, 2019; American Psychological Association, Guideline Development Panel for the Treatment of PTSD in Adults, 2017).

RESPONSE: we do not feel that the evidence base for EMDR is controversial. The clinical evidence for EMDR was derived, as for all interventions, from our systematic review and NMA, described in a related paper cited in the economic manuscript. The NMA paper includes references to other reviews. Nevertheless, we have now included an overview of recommendations on the use of TF-CBT and EMDR in other PTSD clinical practice guidelines in the manuscript, and included reference to the American Psychological Association guidelines (page 34, lines 302-306).

Conclusions/Limitations

7) The manuscripts notes (p. 30) that TF-CBT appeared to be less cost-effective than other interventions, yet had “by far the most solid evidence base.” What implications does that have on the findings? Perhaps TF-CBT yields stronger maintenance of treatment gains over time; although it appears that comparable remission rates were unavailable for most other treatments.

RESPONSE: we have now replaced ‘solid’ by ‘large’ for clarity. A larger evidence base give us more confidence in the findings on clinical effectiveness for TF-CBT; this point has been clarified on page 30, lines 205-206. It is also true that TF-CBT and EMDR were the only interventions with evidence of effect at follow up, and this information has now been added in the manuscript (page 29, lines 185-187).

8) In addition to grouping together TF-CBT by number of sessions, the limitations section of the supplemental manuscript notes that “TF-CBT” includes a broad range of therapies from CPT to mindfulness-based cognitive therapy, whereas other studies (e.g., Gerger et al., 2014) assess CBT, CT, and ET both separately and as a group. Given the high variation among interventions and between-trial heterogeneity, what might be the impact this grouping?

RESPONSE: we have now added text on this issue in our NMA paper, to justify our decision to analyse together different therapies within the TF-CBT class. We have noted that other studies have assessed some of these interventions, for which evidence was adequate, separately; the majority of these studies have found evidence on the effectiveness of all interventions within the TF-CBT class but none of the studies reported any evidence on differential effects between different types of TF-CBT. Gerger et al. found no difference in the effectiveness of different interventions within the TF-CBT class, which supports our decision to consider TF-CBT interventions together, as one class, in our analysis. Moreover, we did an exploratory post-hoc sub-analysis by specific TF-CBT intervention for all studies including a waitlist control, which suggests no significant sub-group difference between specific TF-CBT intervention types. These results are provided in our related clinical [NMA] paper. 

9) How might the timeline and complexity of the trauma histories impact the findings?

RESPONSE: as we have acknowledged in our NMA paper, our NMA analyses were characterised by high between-trial heterogeneity, which may have been caused by heterogeneity across populations included in the trials considered in our analysis, for example, in terms of the presence of a formal PTSD diagnosis, the baseline severity and complexity of PTSD symptoms, the type, extent and multiplicity of trauma exposure, the chronicity of symptoms and the presence of comorbidity. Moreover, the vast majority of the studies included in the NMAs did not distinguish between PTSD and complex PTSD, because our review was undertaken before ICD-11 (and the distinction between PTSD and complex PTSD) was released. Trials are likely to have varied widely in the proportion of participants with complex PTSD and this may have had an impact on the effectiveness of assessed interventions in each study and the heterogeneity across studies.

We have conducted exploratory sub-analyses by multiplicity of index trauma for the TF-CBT versus waitlist comparison for the PTSD symptom change scores between baseline and endpoint outcome; our results showed no significant subgroup differences for single compared to multiple incident index trauma. This information has been added in our NMA paper.

We have also added a paragraph on the factors that may have contributed to the high heterogeneity in the NMA results that informed the economic analysis (page 30, lines 188-198).

10) How do the included studies compare in terms of patient’s treatment histories (e.g., history of multiple psychological treatments and the timing of these treatments)?

RESPONSE: such information (multiplicity and timing of previous psychological treatment) was not reported in most RCTs included in the NMAs, and thus this information was not possible to extract and assess. Nevertheless, we have added discussion on this issue (and its potential impact on heterogeneity) on page 30, lines 194-198.

Overall, this was a well-written manuscript representing an important contribution to PTSD research, while highlighting important gaps in the literature (e.g., long-term follow-ups, applicability of treatments to complex traumas).

RESPONSE: we thank the reviewer for their comment.

Reviewer #3: 

This paper presents results from a detailed and extensive programme of work. It draws on a separately reported strand of analysis on clinical effectiveness of a range of different types of interventions for adults with PTSD. It builds on that work by combining it with new estimations of the costs of each type of intervention when delivered in England, so that both clinical-effectiveness and cost (both to services and to individuals) are considered . The paper presents the methods used to inform the NICE guidelines for PTSD treatment in England: in England NICE guidelines are absolutely pivotal to informing NHS clinical practice and resourcing decisions.

While much of this evidence may exist (in different form) in the NICE documentation alongside the guidelines, it is so important that the research underpinning guidelines is also made available in peer review journal form – for transparency and so it is indexed and identifiable for systematic review. The analysis in this paper also used some different parameters to inclusion to the guidelines. It would be useful if the authors could comment on what impact on results increasing the minimum number of participants in included studies from 50 to 100 had on the overall results (I might have missed this).

RESPONSE: we thank the reviewer for their comment. We have now added this information on page 7, lines 6-10.

Please note I do not have great expertise in clinical cost-effectiveness calculations and do not feel able to properly comment on those methods. The process has been overseen by expert committee (some of whom are also authors), appears detailed, and is well-documented. The figures aid interpretation of methods as well as of findings, esp fig 1.

Such costings inevitably must deal in averages. The decisions on where to anchor those averages seems sensible (although the community data finds rates of screen-positive PTSD to be highest in 16-24 year olds, while the cohort anchors initiation age at 39 to reflect the treatment population. This may well be appropriate. 

RESPONSE: as stated on page 4, lines 10-12, determining the starting age of the cohort served only in estimating mortality of the cohort over time (as mortality changes with age) and had no impact on the effectiveness of the interventions. The community data refer to populations that are screened positive for PTSD, but they do not necessarily seek treatment for this condition. In contrast, as we explain in the manuscript (page 4, lines 6-7), the age of 39 years reflects the mean age of adults with PTSD presenting to healthcare services in the UK, which was the population of interest in our study.

However, in the limitations it would be good to acknowledge that while – overall – EMDR may be the most cost-effective, this could vary with group characteristics. E.g. some interventions may be more clinically effective (and therefore, potentially, also more cost-effective) in the youngest age group, or in those with the most (or least) severe symptoms, or those with comorbid or delayed PTSD. Just something to acknowledge. An understanding of this is important in understanding the applications of the results, and how rigidly the recommendations should be applied in practice. There may be situations where a lower ranked intervention is the most appropriate.

RESPONSE: we have now acknowledged that the NMAs that informed the economic analysis showed high between-study heterogeneity, which is likely to have been caused by heterogeneity across populations included in the trials considered in the NMAs, for example, in terms of the presence of a formal PTSD diagnosis, the severity, complexity and chronicity of PTSD symptoms, the type, extent and multiplicity of trauma exposure, the presence of comorbidity, the variability of interventions within each assessed option and the differences across settings, e.g. inpatient versus outpatient delivery of interventions (page 30, lines 188-198). However, we have no indication that some interventions may be more clinically and cost-effective in people with more or less severe PTSD, or those with comorbid or delayed PTSD. We do have evidence that EMDR is less clinically and cost-effective in younger populations, compared with other treatments, and this information has now been added in the manuscript (page 32, lines 256-263). We also have evidence that EMDR has a non-significant effect in people with combat-related trauma, which has had an impact on NICE recommendations on EMDR for adults with PTSD; this has been reported on page 34, lines 299-300. Furthermore, the NICE recommendations on adults with PTSD acknowledge that lower ranked interventions may be more suitable for some people, hence there are more recommendations on self-help with support, non-TF-CBT and SSRIs; these recommendations have been summarised in the manuscript (page 34, lines 309-318). Ultimately, the choice of treatment should be a joined decision between the clinician and the patient, and this is a wider issue underlying all NICE guidelines and not specific to PTSD guideline recommendations.

I would have liked a line defining each intervention, or at least describing how the interventions evaluated were assigned to the intervention classifications. For example, ‘counselling’ gets a damning verdict – but I am not clear precisely what kinds of interventions were counted here. Are they simply interventions that were so poorly done that they were unclassifiable in any of the other categories? Which could be a confounding factor explaining their poor performance? Were all interventions assigned to one category only, or was there overlap?

RESPONSE: we have now added definitions of all treatment options assessed in the economic analysis on pages 5-6. We can confirm that each intervention was assigned to one category only. 

Abstract has a sentence which reads: ‘TF-CBT has the most solid evidence base’ – take especial care in the abstract (which will be all that many read) that it is clear what this means (the usual disentangling ‘lack of evidence’ and ‘evidence of lack’ is needed).

RESPONSE: we thank the reviewer for their comment. We have now amended ‘solid’ to ‘largest’ for clarity.

---

## [Decision Letter · Decision Letter 1]

10 Mar 2020

PONE-D-19-20536R1

Cost-effectiveness of psychological treatments for post-traumatic stress disorder in adults

PLOS ONE

Dear Dr Mavranezouli,

Thank you for submitting your manuscript to PLOS ONE. After careful consideration, we feel that it has merit but does not fully meet PLOS ONE’s publication criteria as it currently stands. Therefore, we invite you to submit a revised version of the manuscript that addresses the points raised during the review process.

I am enthusiastic about your submission and would like to recommend it for acceptance for publication. Please respond to the reviewer's comments.

We would appreciate receiving your revised manuscript by Apr 24 2020 11:59PM. To enhance the reproducibility of your results, we recommend that if applicable you deposit your laboratory protocols in protocols.io, where a protocol can be assigned its own identifier (DOI) such that it can be cited independently in the future. For instructions see: http://journals.plos.org/plosone/s/submission-guidelines#loc-laboratory-protocols

We look forward to receiving your revised manuscript.

Kind regards,

Scott McDonald

Academic Editor

PLOS ONE

Reviewers' comments:

Reviewer's Responses to Questions

**Comments to the Author**

1. If the authors have adequately addressed your comments raised in a previous round of review and you feel that this manuscript is now acceptable for publication, you may indicate that here to bypass the “Comments to the Author” section, enter your conflict of interest statement in the “Confidential to Editor” section, and submit your "Accept" recommendation.

Reviewer #1: (No Response)

Reviewer #2: All comments have been addressed

2. Is the manuscript technically sound, and do the data support the conclusions?

Reviewer #1: Yes

Reviewer #2: Yes

3. Has the statistical analysis been performed appropriately and rigorously? 

Reviewer #1: Yes

Reviewer #2: Yes

4. Have the authors made all data underlying the findings in their manuscript fully available?

Reviewer #1: Yes

Reviewer #2: Yes

5. Is the manuscript presented in an intelligible fashion and written in standard English?

Reviewer #1: Yes

Reviewer #2: Yes

6. Review Comments to the Author

Reviewer #1: The authors' revisions are responsive and their rebuttals are generally persuasive. A few points of clarification would be helpful. 1. The much lower percentages of studies requiring a PTSD diagnosis evaluating somatic and psychoeducation interventions (25-33%) vs. in EMDR or TF-CBT (75-78%), and the very few studies (3-4) for each of the former modalities, makes conclusions for those modalities quite uncertain. From their NMA, can the authors provide any information (briefly) to help readers determine if the PTSD symptom severity was comparable at baseline for those studies versus for the studies of TF-CBT and EMDR? This would be helpful in judging whether this is really an apples-to-apples comparison of cost effectiveness that should favor somatic and psychoeducation modalities over TF-CBT. 2. The finding that self-help plus support had a relatively strong QUALY outcome and cost-effectiveness, with a relatively large number of studies most of which required a PTSD diagnosis, deserves a bit more highlighting in the Discussion.

Reviewer #2: (No Response)

7. PLOS authors have the option to publish the peer review history of their article (what does this mean?). If published, this will include your full peer review and any attached files.

Reviewer #1: No

Reviewer #2: Yes: Sarah M. Scott, Ph.D

---

## [Author Response · Author response to Decision Letter 1]

3 Apr 2020

Responses to reviewers' comments:

Reviewer #1

The authors' revisions are responsive and their rebuttals are generally persuasive. A few points of clarification would be helpful. 

We thank the reviewer for their useful comments.

1. The much lower percentages of studies requiring a PTSD diagnosis evaluating somatic and psychoeducation interventions (25-33%) vs. in EMDR or TF-CBT (75-78%), and the very few studies (3-4) for each of the former modalities, makes conclusions for those modalities quite uncertain. From their NMA, can the authors provide any information (briefly) to help readers determine if the PTSD symptom severity was comparable at baseline for those studies versus for the studies of TF-CBT and EMDR? This would be helpful in judging whether this is really an apples-to-apples comparison of cost effectiveness that should favor somatic and psychoeducation modalities over TF-CBT.

Response: we agree that the very few studies for somatic and psychoeducation interventions make conclusions for those modalities quite uncertain, and this is reflected in the Discussion of the paper on NICE recommendations:

“Psychoeducation was shown to be cost-effective based on limited and inconclusive clinical evidence; therefore, it was not recommended as a stand-alone intervention, but as part of individual TF-CBT. Finally, the committee noted the evidence of high clinical and cost-effectiveness for combined somatic/cognitive therapies, but also considered their particularly limited evidence base beyond treatment endpoint and the lack of specific indications for these interventions, and decided not to recommend them but instead to make a recommendation for further research” (page 36, lines 365-371).

Regarding inclusion criteria on PTSD diagnosis versus clinically important PTSD symptoms: We have already acknowledged this issue as a potential factor contributing to the high heterogeneity of the NMA: “Both NMAs were characterised by high between-trial heterogeneity, which is likely to have been caused by heterogeneity across populations included in the trials considered in the NMAs, for example, in terms of the presence of a formal PTSD diagnosis, the severity, complexity and chronicity of PTSD symptoms…” (page 30, lines 187-190).

Unfortunately it was not possible to determine if severity was comparable across trials at baseline using the mean PTSD symptom severity scores, because the studies have used a range of different PTSD symptom scales and no mapping function that could allow comparison of severity levels across the scales is available. We have attempted to assess comparability of populations by looking at the % of the mean score over cut-off or the % of mean score out of the total score across studies, but we concluded that this was not reliable because it is expected that different symptom scales will have different levels of redundancy built in.

However, we note that the categorisation of diagnosis versus clinically important symptoms is based on inclusion criteria so it is possible that most of the participants in the ‘clinically important symptom’ studies could have a diagnosis, they were just not required to do so in order to participate in the trial, and this could well have been for pragmatic reasons to do with trial management rather than an indication of differing severity. It is also the case that for scales that are based closely on diagnostic criteria, e.g. PTSD Checklist (PCL), scoring above the clinical threshold may be regarded as comparable to receiving a diagnosis. The studies that evaluate combined somatic and cognitive therapies all use scales that are based on diagnostic criteria (e.g. PCL and Modified PTSD Symptom Scale [MPSS-SR]), and the majority of the studies evaluating psychoeducation also use scales based on diagnostic criteria (e.g. PCL and Davidson Trauma Scale).

We have now added the above points in the Discussion section (page 30, lines 199 to page 31, line 235).

2. The finding that self-help plus support had a relatively strong QUALY outcome and cost-effectiveness, with a relatively large number of studies most of which required a PTSD diagnosis, deserves a bit more highlighting in the Discussion.

Response: we have now added discussion on the cost-effectiveness of self-help with support (page 36, lines 350-58). We have covered discussion on the issue of formal diagnosis of PTSD vs clinically important PTSD symptoms and the potential impact on the comparability of populations regarding baseline symptom severity – please refer to our response to your previous comment.

---

## [Editor Report · Decision Letter 2]

13 Apr 2020

Cost-effectiveness of psychological treatments for post-traumatic stress disorder in adults

PONE-D-19-20536R2

Dear Dr. Mavranezouli,

We are pleased to inform you that your manuscript has been judged scientifically suitable for publication and will be formally accepted for publication once it complies with all outstanding technical requirements.

With kind regards,

Scott McDonald

Academic Editor

PLOS ONE
---

## [Editor Report · Acceptance letter]

20 Apr 2020

PONE-D-19-20536R2 

Cost-effectiveness of psychological treatments for post-traumatic stress disorder in adults 

Dear Dr. Mavranezouli:

I am pleased to inform you that your manuscript has been deemed suitable for publication in PLOS ONE. Congratulations! Your manuscript is now with our production department. 

With kind regards,

on behalf of

Dr. Scott McDonald 

Academic Editor

PLOS ONE